# TWO-SIDED COMPETING MATCHING MARKETS WITH COMPLEMENTARY PREFERENCES

## ABSTRACT

In this paper, we propose a new algorithm for addressing the problem of two-sided matching markets with complementary preferences, where agents' preferences are unknown a priori and must be learned from data. The presence of complementary preferences can lead to instability in the matching process, making this problem challenging to solve. To overcome this challenge, we formulate the problem as a bandit learning framework and propose the Multi-agent Multi-type Thompson Sampling (MMTS) algorithm. The algorithm combines the strengths of Thompson Sampling for exploration with a double matching technique to achieve a stable matching outcome. Our theoretical analysis demonstrates the effectiveness of MMTS as it is able to achieve stability at every matching step and has a sublinear Bayesian regret over time. Our approach provides a useful method for addressing complementary preferences in real-world scenarios.

## 1 INTRODUCTION

Two-sided matching markets have been a mainstay of theoretical research and real-world applications for several decades since the seminal work by (Gale and Shapley, 1962). Matching markets are used to allocate indivisible "goods" to multiple decision-making agents based on mutual compatibility as assessed via sets of preferences. Matching markets embody a notion of scarcity in which the resources on both sides of the market are limited. One of the key concepts that contribute to the success of matching markets is *stability*, which criterion ensures that all participants have no incentive to block a prescribed matching (Roth, 1982). Matching markets often consist of participants with *complementary* preferences that can lead to instability (Che et al., 2019). Examples of complementary preferences in matching markets include: firms seeking workers with skills that complement their existing workforce, sports teams forming teams with players that have complementary roles, and colleges admitting students with diverse backgrounds and demographics that complement each other. Studying the stability issue in the context of complementary preferences is crucial in ensuring the successful functioning of matching markets with complementarities.

In this paper, we propose a novel algorithm and present an in-depth analysis of the problem of complementary preferences in matching markets. Specifically, we focus on a many-to-one matching scenario and use the job market as the example. In our proposed model, there are a set of agents (e.g., firms), each with limited quota, and a set of arms (e.g., workers), each of which can be matched to at most one agent. Each arm belongs to a unique type, and each agent wants to match with a minimum quota of arms from each type. This leads to complementarities in agents' preferences. Additionally, the agents' preference of arms from each type is unknown a priori and must be learned from data, which we refer to as the problem of *competing matching under complementary preference* (CMCP).

Our first result is the formulation of CMCP into a bandit learning framework as described in (Lattimore and Szepesvári, 2020). Using this framework, we propose a new algorithm, the Multi-agent Multi-type Thompson Sampling (MMTS), to solve CMCP. Our algorithm builds on the strengths of Thompson Sampling (TS) in terms of exploration and further enhances it by incorporating a *double matching* technique to find a stable solution under CMCP. The TS algorithm, as described in (Thompson, 1933; Agrawal and Goyal, 2012; Russo et al., 2018), can effectively address the *incapable exploration* problem in the competing matching problem, as described in (Liu et al., 2020), by using the randomized sampling, also illustrated in Section 3.2. Unlike the upper confidence bound (UCB) algorithm, TS method can achieve sufficient exploration by incorporating a deterministic,

non-negative bias inversely proportional to the number of matches into the observed empirical means. Furthermore, the double matching technique proposed in this paper uses two stages of matching to satisfy both the type quota and total quota requirements. These two stages mainly consist of using the deferred acceptance (DA) algorithm from (Gale and Shapley, 1962), which is easy to be implemented.

Second, we present a theoretical analysis of the proposed MMTS algorithm. Our analysis shows that MMTS can achieve stability at each matching step and show the incentive compatibility (IC) of the MMTS. The proof of stability is obtained through a two-stage design of the *double matching* technique, and the proof of IC is obtained through the lower bound of the regret. To the best of our knowledge, MMTS is the first algorithm to achieve stability and IC in the CMCP.

Finally, our theoretical results indicate that MMTS can achieve a Bayesian total regret that scales with the square root of the time horizon ($T$) and is nearly linear in the total quota of all firms ($Q$). Furthermore, we find that the Bayesian total regret only depends on the square root of the *maximum number of workers* ($K_{\max}$) in one type rather than the square root of the total number of workers ($\sum_m K_m$) in all types. This is a more challenging setting than that considered in previous works such as (Liu et al., 2020; Jagadeesan et al., 2021), which only consider a single type of worker in the market and a quota of one for each firm. To address these challenges, we use the eluder dimension (Russo and Van Roy, 2013) to measure the uncertainty set widths and bound the instantaneous regret for each firm, and use the union bound of concentration results to measure the probability of *bad events* occurring to get the final regret. Bounding the uncertainty set width is the key step for deriving the sublinear regret upper bound of MMTS.

The rest of this paper is organized as follows. In Section 2, we introduce the necessary components in the problem of CMCP. Meanwhile, we also state the challenges of this problem. In Section 3, we provide the MMTS algorithm, its comparison with other algorithms, and show the incapability of the UCB algorithm in CMCP. Then we present the stability, regret upper bound, and the incentive-compatibility of the MMTS in Section 4. Finally, in Section 5, we show two examples, present the distribution of learning parameters, and demonstrate the robustness of MMTS in large markets.

## 2 PROBLEM

### 2.1 PROBLEM FORMULATION

We now describe the problem formulation of the **C**ompeting **M**atching under **C**omplementary **P**references problem (CMCP). Using the scenario of worker-firm matching as our running example, we introduce the notation and key components of the CMCP. We define $T$ as the time horizon and without loss of generality, we assume it is known[1]. We denote $[N] = [1, 2, ..., N]$ where $N \in \mathbb{N}^+$. Define the bold $\mathbf{x} \in \mathbb{R}^d$ be a $d$-dimensional random vector.

**(I) Environment.** We consider a centralized platform with $N$ firms, denoted by the set $\mathcal{N} = \{p_1, p_2, ..., p_N\}$, and various types of workers, represented by sets $\mathcal{K}_m = \{a_1^m, a_2^m, ...a_{K_m}^m\}$, for $m \in [M]$, where $K_m$ is the number of $m$-th type workers. Each firm $p_i$ has a specific minimum type quota $q_i^m$ to recruit $m$-type workers, and a maximum total quota $Q_i$ (e.g., seasonal headcount in company), for all $i \in [N], m \in [M]$ and we assume $\sum_{i=1}^M q_i^m \leq Q_i$. Additionally, we define the total market quota as $Q = \sum_{i=1}^N Q_i$ and the total number of available market workers as $K = \sum_{m=1}^M K_m$. It is assumed without loss of generality that the total number of quotas is greater less than the total number of available market workers ($Q \ll K$) and $T$ is large.

**(II) Preference.** We give preferences of both sides of the market. There are two preference lists: the preferences of workers towards firms, and the preferences of firms towards workers.

*a. Preferences of $m$-type workers towards firms $\boldsymbol{\pi}^m : \mathcal{K}_m \mapsto \mathcal{N}, \forall m \in [M]$.* We assume that preferences of types of worker to firms are fixed, known over time. For instance, workers submit their preferences over different firms to the platform. We denote $\pi_{j,i}^m$ as the rank order of firm $p_i$ in the preference list of $m-$ type worker $a_j^m$, and assume that there are no ties in the rank orders[2]. The centralized platform knows the fixed preferences of $m-$ type worker towards firms, denoted as

---

[1]The unknown $T$ can be handled with the well-known doubling trick (Auer et al., 1995).

[2]The strict preference is not necessary for stable matching and regret metric. However, for simplicity, we assume preference is strict, which avoids the multiple optimal stable matching solutions even they are equivalent.

$\boldsymbol{\pi}_j^m \subseteq \{\pi_{j,1}^m, ..., \pi_{j,N}^m\} \cup \{a_j\}, \forall a_j \in \mathcal{K}_m$ and $m \in [M]$, and singleton $a_j$ represents the worker's preference to remain unmatched. In other words, $\boldsymbol{\pi}_j^m$ is a subset of the permutation of $[N]$ plus the worker itself. And $\pi_{j,i}^m < \pi_{j,i'}^m$ implies that $m - type$ worker $a_j^m$ prefers firm $p_i$ over firm $p_{i'}$ and as a shorthand, denoted as $p_i <_j^m p_{i'}$. This known worker-to-firm preference is a mild and common assumption in the matching market literature (Liu et al., 2020; 2021; Li et al., 2022).

*b. Preferences of firms towards $m - type$ workers* $\mathbf{r}^m : \mathcal{N} \mapsto \mathcal{K}_m, \forall m \in [M]$. The true *unknown* preferences of firms towards workers are fixed over time. The goal of the platform is to infer these unknown preferences through historical matching data. We denote $r_{i,j}^m$ as the true rank of worker $a_j^m$ in the preference list of firm $p_i$, and assume there are no ties. $p_i$'s preferences towards workers is represented by $\mathbf{r}_i^m$, which is a subset of the permutation of $[\mathcal{K}_m]$ plus the firm $p_i$ itself, representing the firm's preference to remain unmatched. Here $r_{i,j}^m < r_{i,j'}^m$ implies that firm $p_i$ prefers worker $a_j^m$ over worker $a_{j'}^m$, and the notation $a_j^m <_i^m a_{j'}^m$ similarly denotes this preference.

We refer to the above preference setting as *marginal preferences* (MP). To illustrate the distinction between marginal preferences and joint (couple) preferences (JP) (Che et al., 2019), we provide an example in Figure 1 involving two types of workers as an example. In the MP setting, preferences of each type of worker are independent of those of the other types. Here we use $x$-axis to represent the firm $p_i$'s preference levels or called utility ($\mu_i^1 \in [0, 1]$) over type 1 workers ($a_1, a_2$), shown as red circles, and we use the $y$-axis to represent the firm $p_i$'s preference levels ($\mu_i^2 \in [0, 1]$) over type 2 workers ($b_1, b_2$), shown as blue triangles. The first row of Figure 1 represents the MP setting, while the second row represents *the JP that the MP cannot cover*.

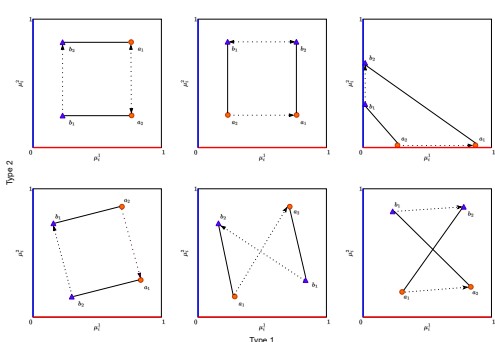

Figure 1: MP v.s. JP.

In the MP setting, we show three possible patterns of combination matchings, $\{(a_1, b_2), (a_2, b_1)\}$ where two types of workers $a$ and $b$ are matched by firm, concatenated a solid line. Besides, a dashed arrow ($a_1 \to a_2$) is used to represent the preference over matching, indicating that $a_1 <_i a_2$. Thus, the first row can be captured by the MP setting as the preference is consistent across the two types of workers. In contrast, the second row cannot be represented by the MP as it is not possible to compare the preference between $(a_1, b_2)$ and $(a_2, b_1)$ using $\{(a_1 > a_2), (b_2 < b_1)\}$. This MP setting is similar to the responsive preferences as defined in (Roth, 1985). Here we only consider the MP and the finding of efficient algorithm to solve the JP problem is notoriously difficult and unsolved (Che et al., 2019) and deserved more efforts.

**(III) Matching Policy.** At time $t$, for each firm $p_i$, $u_t^m(p_i) : \mathcal{N} \mapsto \mathcal{K}_m \cup \emptyset$ is a mapping function that satisfies $u_t^m(p_i) \in \mathcal{K}_m \cup \emptyset, \forall i \in [N]$, where $\emptyset$ represents a null set. At each time $t$, the platform assigns $m - type$ workers $u_t^m(p_i)$ for firm $p_i$. We define $u_t^m(p_i)$ as the function that maps each firm $p_i$ in the set $\mathcal{N}$ to a set of $m - type$ workers $\mathcal{K}_m$ at time $t$. The assignments $u_t^m(p_i)$ for each firm are not only based on the firm's proposals (submitted preferences), but also on the *competing status* with other firms and workers' preferences. A centralized platform, such as LinkedIn or Amazon Mechanical Turks, coordinates this matching process in this competitive environment.

**(IV) Stable Matching.** The concept of *stability* is a widely used notion in the literature of stable matching, which refers to the property that no pair of agents (e.g., firms and workers) would mutually prefer each other over their current match (Gale and Shapley, 1962; Roth, 2008). This property is typically formalized as the absence of *blocking pairs* in the matching literature, which are pairs of agents that would both prefer to be matched with each other over their current match. The formal definition is illustrated as below.

**Definition 1.** *(Blocking pair). A matching $u$ is blocked by a firm $p_i$ if $p_i$ prefers being single to being matched with $u(p_i)$, i.e. $p_i >_i u(p_i)$. A matching $u$ is blocked by a pair of firm and worker $(p_i, a_j)$ if they each prefer each other to the partner they receive at $u$, i.e. $a_j >_i u(p_i)$ and $p_i >_j u^{-1}(a_j)$.*

---

Roth (2008) stated if some preferences are not strict, arbitrarily breaking ties lets each agent fill out a strict preference list.

**Definition 2.** *(Stable Matching). A matching* $u$ *is stable if it isn't blocked by any individual or pair of worker and firm.*

In this setting, however, each firm has a minimum quota vector $\mathbf{q}_i = [q_i^1, ..., q_i^M] \in \mathbb{R}^M$ for each type of worker to fill. Therefore, we define the concept of *stability* as the absence of "blocking pairs" across all types of workers and firms. Based on the definition of the stable matching, we also discussed the feasibility of the stable matching in the Appendix A. Here without loss of generality, we assume there exists the stable matching in the scheme of the complementary preference.

**(V) Matching Reward.** At time $t$, when firm $p_i$ is matched with worker $a_j^m$, the firm receives a stochastic reward $y_{i,j}^m(t)$ which is assumed to be the *true matching reward* $\mu_{i,j}^m(t)$ plus a noise $\epsilon_{i,j}^m(t)$,

$$y_{i,j}^m(t) = \mu_{i,j}^m(t) + \epsilon_{i,j}^m(t), \forall i \in [N], \forall j \in [K_m], \forall m \in [M], \forall t \in [T], \tag{1}$$

where we assume that $\epsilon_{i,j}^m(t)$'s are independently drawn from a sub-Gaussian random variable with parameter $\sigma$. That is, for every $\alpha \in \mathbb{R}$, it is satisfied that $\mathbb{E}[\exp(\alpha \epsilon_{i,j}^m(t))] \leq \exp(\alpha^2 \sigma^2 / 2)$. The goal of the centralized platform is to design a learning algorithm that achieves stable matchings through learning the firms' preferences for multiple types of workers preciously from the previous matchings.

**(VI) Regret.** Based on model (1), we can observe a matching reward $\mathbf{y}_i^m(t) := \mathbf{y}_{i,u_t^m(p_i)}(t)$ at time $t$ when firm $p_i$ is matched with the assigned $m-$type workers $u_t^m(p_i)$. The mean reward $\mu_{i,u_t^m(p_i)}(t)$ represents the noiseless reward (or mean utility) of firm $p_i$ wrt its assigned matching $m-$type workers $\mathbf{u}_t^m(p_i)$ at time $t$. We define the cumulative *firm-optimal regret with $m$-type worker* for firm $p_i$ as

$$R_i^m(T, \theta) := \sum_{t=1}^{T} \mu_{i,\overline{u}_i^m} - \sum_{t=1}^{T} \mu_{i,u_t^m(p_i)}(t), \tag{2}$$

where we denote $\theta$ as the sampled problem instance, and it is independently generated from a distribution $\Theta$. This firm-optimal regret represents the difference between the capability of a policy $u_i^m := \{u_t^m(p_i)\}_{t=1}^T$ in hindsight and the optimal stable matching *oracle policy* $\overline{u}_i^m$. As each firm must recruit $M$ types of workers with total quota $Q_i$, the *total cumulative firm-optimal stable regret* for firm $p_i$ is defined as the sum of this difference over all types of workers, $R_i(T, \theta) := \mathbb{E}\left[\sum_{m=1}^{M} \sum_{t=1}^{T} \mu_{i,\overline{u}_i^m} - \sum_{m=1}^{M} \sum_{t=1}^{T} \mu_{i,u_t^m(p_i)}(t) | \theta\right]$. Finally, the *Bayesian total cumulative firm-optimal stable regret* for all firms is defined as the expected value of the total cumulative firm-optimal stable regret over all firms, $\mathfrak{R}(T) := \mathbb{E}_{\theta \in \Theta}\left[\sum_{i=1}^{N} R_i(T, \theta)\right]$. Our goal is to design an algorithm that minimizes this value over the time horizon $T$.

## 2.2 Challenges and Solutions

When preferences are unknown a priori in matching markets, the stability issue while satisfying complementary preferences and quota requirements is a challenging problem due to the interplay of multiple factors.

**Challenge 1: How to design a stable matching algorithm for markets with complementary preferences?** This is a prevalent issue in real-world applications such as hiring workers with complementary skills in hospitals and high-tech firms or admitting students with diverse backgrounds in college admissions. Despite its importance, no implementable algorithm is currently available to solve this challenge. In this paper, we propose a novel approach to resolving this issue by utilizing a *double matching* (Algorithm 3) to *marginalize* complementary preferences and achieve stability. Our algorithm can efficiently learn a stable matching solution using historical matching data, providing a practical solution to the problem of competing matching under complementary preferences.

**Challenge 2: How to balance the exploration and exploitation to achieve the sublinear regret?** The centralized platform must find a way to collect more firm-worker matching feedback while also achieving optimal matching at each time step. Compared to traditional matching algorithms, the CMCP is more challenging as it requires more time to balance this trade-off. In previous research, the classic UCB method could not achieve sublinear regret in some scenarios (Liu et al., 2020). We will also show an example in Section 3.2 to illustrate it. To overcome this challenge, we propose the use of TS algorithm which allows for random exploration and achieves sublinear regret.

---

**Algorithm 1:** Multi-agent Multi-type Thompson Sampling (MMTS)

---

**Input** : Time horizon $T$; firms' priors $(\boldsymbol{\alpha}_i^{m,0}, \boldsymbol{\beta}_i^{m,0}), \forall i \in [N], \forall m \in [M]$; workers' preference $\boldsymbol{\pi}^m, \forall m \in [M]$.

1 **for** $t \in \{1, ..., T\}$ **do**
2     STEP 1: SAMPLING STAGE
3        Get all firms' estimated rankings $\hat{\mathbf{r}}_i^m(t)$ and estimated mean reward $\hat{\boldsymbol{\mu}}_i^m(t)$ over all types of workers, $\forall i \in [N], m \in [M]$ from the Sampling stage in Algo 2.
4     STEP 2: DOUBLE MATCHING STAGE
5        Get the matching result $\mathbf{u}_t^m(p_i), \forall i \in [N], m \in [M]$ from the *double matching* in Algo 3.
6     STEP 3: COLLECTING REWARDS STAGE
7        Each firm receives its corresponding rewards from all types of workers $\mathbf{y}_i^m(t)$.
8     STEP 4: UPDATING BELIEF STAGE
9        Based on received rewards, firms update their posterior belief.

---

**Challenge 3: How to solving CMCP with quota constraints in large markets?** Unlike the classic DA algorithm (Gale and Shapley, 1962), our problem involves type-specific and total quota requirements for each firm. Can we find a stable matching algorithm that satisfies these constraints while also adapting to unknown preferences? Furthermore, can this algorithm be applied in large markets with efficiency? We address these challenges by proposing a novel algorithm, double matching, that effectively balances exploration and exploitation while can also be partially parallel implemented.

## 3 ALGORITHMS

In this section, we propose the Multi-agent Multi-type Thompson Sampling algorithm (MMTS), which aims to learn the true preferences of all firms over all types of workers, achieve stable matchings, and maximize the firms' Bayesian expected reward. We provide a detailed description of the algorithm and demonstrate its benefits of using TS. Besides, we also discuss the computational complexity of MMTS in Appendix B.

### 3.1 ALGORITHM DESCRIPTION

The MMTS in Algorithm 1 is composed of four stages, *sampling stage*, *double matching stage*, *collecting reward stage*, and *updating belief stage*. The common knowledge for centralized platform is the time horizon $T$, the number of participants, workers' preference $\{\boldsymbol{\pi}^m\}_{m=1}^M$, and firms' learning priors $\{(\boldsymbol{\alpha}_i^{m,0}, \boldsymbol{\beta}_i^{m,0})\}_{m=1}^M, \forall i \in [N]$. Then at each matching step $t$, MMTS iterates these four steps.

**Step 1: Sampling Stage.** For each firm $p_i$, it samples the estimated mean reward $\hat{\mu}_{i,j}^m(t)$ for $m-$ type worker $a_j^m$ from a specific distribution $\mathcal{P}_j^m$ (e.g., Gaussian or beta distribution) with learned parameters $(\alpha_{i,j}^{m,t-1}, \beta_{i,j}^{m,t-1})$ from the previous time step $t-1$, which is $\hat{\mu}_{i,j}^m(t) \sim \mathcal{P}(\alpha_{i,j}^{m,t-1}, \beta_{i,j}^{m,t-1}), \forall i \in [N], \forall m \in [M], \forall j \in [\mathcal{K}_m]$. Besides, for the firm $p_i$, it sorts these type-specific workers based on the sampled mean reward $\{\hat{\mu}_{i,j}^m(t)\}_{j=1}^{K_m}$ in descending order and gets the estimated rank $\hat{\mathbf{r}}_i^m(t)$ for $m-$ type workers. After that, all firms submit their estimated ranks to the centralized platform. The above steps are shown in Algorithm 2.

**Step 2: Double matching stage.** With the shared estimated mean rewards $\hat{\boldsymbol{\mu}}(t) := \{\hat{\mu}_{i,j}^m(t)\}_{i,j,m}$ and estimated ranks $\hat{\mathbf{r}}(t) := \{\hat{\mathbf{r}}_i^m(t)\}_{i,m}$ from firms at time $t$, the double matching algorithm takes these ranks and quota constraints from firms as input and match firms and workers in two-stage matchings as shown in Algorithm 3. The goal of the first match is to allow all firms to satisfy their minimum type-specific quota $q_i^m$ first. The second match is to fill the left-over positions $\tilde{Q}_i$ (defined below) for each firm and match firms and workers without type consideration. Before implementing the second match, we have to sanitize the status quo as a priori.

*First Match:* The platform implements the type-specific DA in Appendix Algorithm 4 given quota requirement $\{q_i^m\}_{m=1}^M, \forall i \in [N]$. The matching road map starts from matching all firms with type from 1 to $M$ and returns the matching result $\{\tilde{u}_t^m(p_i)\}_{m \in [M]}$, which can be implemented in parallel.

---

**Algorithm 2:** Sampling Stage

---

**Input** : Time horizon $T$; firms' priors $(\boldsymbol{\alpha}_i^{m,0}, \boldsymbol{\beta}_i^{m,0}), \forall i \in [N], \forall m \in [M]$.
1 **Sample**: Sample mean reward $\hat{\mu}_{i,j}^m(t) \sim \mathcal{P}(\alpha_{i,j}^{m,t-1}, \beta_{i,j}^{m,t-1}), \forall i \in [N], \forall m \in [M], \forall j \in [\mathcal{K}_m]$.
2 **Sort**: Sort estimated mean rewards $\hat{\mu}_{i,j}^m(t)$ in descending order and get the estimated rank $\hat{\mathbf{r}}_i^m(t)$.
3 **Output**: The estimated rank $\hat{\mathbf{r}}_i^m(t)$ and the estimated mean rewards $\hat{\boldsymbol{\mu}}_i^m(t), \forall i \in [N], m \in [M]$.

---

**Algorithm 3:** Double Matching

---

**Input** : firms' estimated rank $\hat{\mathbf{r}}_i^m(t)$, estimated mean $\hat{\boldsymbol{\mu}}_i^m(t)$, type quota $q_i^m, \forall m \in [M], i \in [N]$
and total quota $Q_i, \forall i \in [N]$; workers' preference $\{\boldsymbol{\pi}^m\}_{m \in [M]}$.
1 STEP 1: FIRST MATCH
2     Submit all firms' estimated ranks $\hat{\mathbf{r}}_i^m(t)$ and all workers' preferences $\boldsymbol{\pi}^m$ to the platform.
3     Run the firm choice DA in Algo 4 and return the matching $\tilde{u}_t^m(p_i)$ for firms over all types.
4 STEP 2: SANITIZE QUOTA
5     Sanitize whether all firms' positions have been filled. For each company $p_i$, if
$Q_i - \sum_{m=1}^M q_i^m > 0$, set the left quota as $\tilde{Q}_i \leftarrow Q_i - \sum_{m=1}^M q_i^m$ for firm $p_i$.
6 STEP 3: SECOND MATCH
7 **if** $\tilde{\mathbf{Q}} \neq 0$ **then**
8 |    Submit left quota $\{\tilde{Q}_i\}_{i \in [N]}$, estimated means $\hat{\boldsymbol{\mu}}(t)$, and workers' preferences $\{\boldsymbol{\pi}^m\}_{m \in [M]}$
|    to the centralized platform. Run the firm choice DA Algo 5 and return the matching $\breve{u}_t(p_i)$.
9 **else**
10 |    Set the matching $\breve{u}_t(p_i) = \emptyset$.
**Output** : The matching $u_t^m(p_i) \leftarrow \text{Merge}(\tilde{u}_t^m(p_i), \breve{u}_t(p_i))$ for all firms.

---

*Sanitize Quota:* After Step 1's first match, the centralized platform will sanitize each firm's left-over quota $\tilde{Q}_i = Q_i - \sum_{m=1}^M q_i^m, \forall i \in [N]$. If there exists a firm $p_i, s.t., \tilde{Q}_i > 0$, then the platform will step into the second match. For those firms like $p_i$ whose leftover quota is zero $\tilde{Q}_i = 0$, their matched workers will skip the second match.

*Second Match:* When rest firms and workers continue to join in the second match, the centralized platform implements the standard DA in Algorithm 5 without type consideration. That is, each firm will re-rank the rest $M$ types of workers who do not have a match in the first match, and fill available vacant positions. It is worth noting that in Algorithm 5, each firm will not propose to the previous workers who rejected him/her and already matched worked in Step 1. Then firm $p_i$ gets the corresponding matched workers $\breve{u}_t(p_i)$ in the second match. Finally, the centralized platform merges the first and second results to obtain a final matching for firm $p_i$ with $m-$ type worker $\mathbf{u}_t^m(p_i) = \text{Merge}(\tilde{u}_t^m(p_i), \breve{u}_t(p_i)), \forall i \in [N], m \in [M]$ at time step $t$.

**Step 3: Collecting Rewards Stage.** When the platform broadcasts the matching result $\mathbf{u}_t^m(p_i)$ to all firms, each firm then receives its corresponding stochastic reward $\mathbf{y}_i^m(t), \forall i \in [N], m \in [M]$.

**Step 4: Updating Belief Stage.** After receiving these noisy rewards, firms update their belief (posterior) parameters as follows, $(\boldsymbol{\alpha}_i^{m,t}, \boldsymbol{\beta}_i^{m,t}) = \text{Update}(\boldsymbol{\alpha}_i^{m,t-1}, \boldsymbol{\beta}_i^{m,t-1}, \mathbf{y}_i^m(t)), \forall i \in [N], \forall m \in [M]$.

In summary, Algorithm 2 samples mean rewards and ranks based on historical matching data. Algorithm 3 computes the stable matching based on the estimated rewards and ranks from the sampling step. Step 3 (collecting rewards) and Step 4 (updating belief) update learning parameters based on received feedback from the assigned matching.

## 3.2 INCAPABLE EXPLORATION

We show why the TS has an advantage over the UCB style method in estimating the ranks of workers. We even find that centralized UCB does achieve linear firm-optimal stable regret in some cases and show it in Appendix C with detailed experimental setting and analysis.Why TS is capable of avoiding the curse of linear regret? By the property of sampling shown in Algorithm 2. Firm $p_i$'s initial prior over worker $a_i$ is a uniform random variable, and thus $r_j(t) > r_i(t)$ with probability $\hat{\mu}_j \approx \mu_j$,

rather than *zero*! This differs from the UCB style method, which cannot update $a_i$'s upper bound due to lacking exploration over $a_i$. The benefit of TS is that it can occasionally explore different ranking patterns, especially when there exists such a previous example. In Figure 2(a), we show a quick comparison of centralized UCB (Liu et al., 2020) in the settings shown above and MMTS when $M = 1, Q = 1, N = 3, K = 3$. The UCB method occurs a linear regret for firm 1 and firm 2. However, TS method suffers a sublinear regret in firm 1 and firm 2.

## 4 PROPERTIES OF MMTS: STABILITY AND REGRET

In Section 4.1, we demonstrated the double matching technique providing the stability property for CMCP. Then we established the Bayesian regret upper bound for all firms when they follow the MMTS in Section 4.2. And we discussed the incentive-compatibility of the MMTS in Appendix G.

### 4.1 STABILITY

In the following theorem, we show the double matching technique can provide stable matching solution based on preferences from the firms' preferences over multiple types of workers provided by the MMTS and fixed and known preferences from workers.

**Theorem 4.1.** *Given two sides' strict preferences from firms and $M$ types of workers. The double-matching procedure can provide a firm-optimal stable matching solution $\forall t \in [T]$.*

*Proof.* The detailed proof can be found in Appendix Section E. □

*Remark.* The sketch proof of the stability property of MMTS is two steps, naturally following the design of MMTS. The first match is conducted in parallel, and the output is stable and guaranteed by (Gale and Shapley, 1962). As the need of MMTS, before the second match, firms without leftover quotas ($\tilde{Q} = 0$) will quit the second round of matching, which will not affect the stability. After the quota sanitizing stage, firms and leftover workers will continue to join in the second matching stage, where firms do no need to consider the type of workers designed by double matching. And the standard DA algorithm will provide a stable result based on each firm's *sub-preference* list. The reason is that for firm $p_i$, all previous possible favorite workers have been proposed in the first match. If they are matched in the first match, they quit together, which won't affect the stability property; otherwise, the worker has a better candidate (firm) and has already rejected the firm $p_i$. So for each firm $p_i$, it only needs to consider a sub-preference list excluding the already matched workers in the first match and the proposed workers in the first match. It will provide a stable match in the second match and won't be affected by the first match. So the overall double matching is a stable algorithm.

### 4.2 BAYESIAN REGRET UPPER BOUND

Next we provide the MMTS algorithm's Bayesian total cumulative firm-optimal regret upper bound.

**Theorem 4.2.** *Assume $K_{\max} = \max\{K_1, ..., K_M\}, K = \sum_{m=1}^{M} K_m$, with probability $1 - 1/QT$, when all firms follow the MMTS algorithm, firms together will suffer the Bayesian expected regret $\mathfrak{R}(T) \leq 8Q \log(QT)\sqrt{K_{\max}T} + NK/Q$.*

*Proof.* The detailed proof can be found in Appendix F. □

*Remark.* The derived Bayesian regret bound, which is dependent on the square root of the time horizon $T$ and a logarithmic term, is nearly rate-optimal. Additionally, we examine the dependence of this regret bound on other key parameters. The first of which is a near-linear dependency on the total quota $Q$. Secondly, the regret bound is dependent only on the *square root* of the maximum worker $K_{\max}$ of one type, as opposed to the total number of workers, $\sum_{m=1}^{M} K_m$ in previous literature (Liu et al., 2020; Jagadeesan et al., 2021). This highlights the ability of our proposed algorithm, MMTS, to effectively capture the interactions of multiple types of matching in CMCP. The second term in the regret is a constant which is only dependent on constants $N, K$ and the total quota $Q$. Notably, if we assume that each $q_i = 1$ and $Q_i = M$, then $NK/Q$ will be reduced to $NK/(NM) = K/M$, which is an unavoidable regret term due to the exploration in bandits (Lattimore and Szepesvári, 2020). This also demonstrates that the Bayesian total cumulative firm-optimal exploration regret is only dependent on the *average* number of workers of each type available in the market, as opposed to the *total* number of workers or the maximum number of workers available of all types. Additionally, if one $Q_i$ is dominant over other firms' $Q_i$, then the regret will mainly be determined by that dominant quota $Q_i$ and $K_{\max}$, highlighting the inter-dependence of this complementary matching problem.

## 5 EXPERIMENTS

In this section, we present simulation results to demonstrate the effectiveness of MMTS in learning the unknown preferences of firms. The overall experiment setup can be found in Appendix I. In Section 5.1, we present two examples and analyze the underlying causes of the interesting phenomenon of negative regret. In Appendix I.2, we showcase the learning parameters from MMTS and provide insight into the reasons for non-optimal stable matchings. Additionally, we demonstrate the robustness of MMTS in large markets in Appendix I.3. All simulation results are run in 100 trials.

### 5.1 TWO EXAMPLES

**Example 1.** There are $N = 2$ firms, $M = 2$ types of workers, and there are $K_m = 5$ free workers for each type. The quota $q_i^m$ for each type and each firm $p_i$ is 2, and the total quota/capacity for each firm is $Q_i = 5$. The time horizon is $T = 2000$.

**Preferences.** True preferences from all types of workers to firms and from firms to different types of workers are all randomly generated. Workers to firms' preferences $\{\boldsymbol{\pi}^m\}_{m=1}^M$ are fixed and known. We use the data scientist (*D/DS*) and software developer engineer (*S/SDE*) as our example. The following are randomly generated true preferences for two-sided participants,

$$
\begin{aligned}
& D_1 : p_1 \succ p_2, \quad D_2 : p_1 \succ p_2, \quad D_3 : p_2 \succ p_1, \quad D_4 : p_1 \succ p_2, \quad D_5 : p_2 \succ p_1, \\
& S_1 : p_1 \succ p_2, \quad S_2 : p_1 \succ p_2, \quad S_3 : p_2 \succ p_1, \quad S_4 : p_2 \succ p_1, \quad S_5 : p_1 \succ p_2, \\
& \pi_1^1 : D_4 \succ D_2 \succ D_3 \succ D_5 \succ D_1, \quad \pi_1^2 : S_1 \succ S_4 \succ S_5 \succ S_2 \succ S_3, \\
& \pi_2^1 : D_2 \succ D_3 \succ D_1 \succ D_5 \succ D_4, \quad \pi_2^2 : S_4 \succ S_2 \succ S_5 \succ S_1 \succ S_3.
\end{aligned}
\tag{3}
$$

The true matching reward of each worker for the firm is randomly generated from the uniform distribution $U([0,1])$, and shown in Appendix Table 1. In addition, noisy reward $y_{i,j}^m(t)$ received (0 or 1) by each firm is generated by the Bernoulli distribution $y_{i,j}^m(t) \sim \text{Ber}(\mu_{i,j}^m(t))$, where $\mu_{i,j}^m(t)$ is the true matching reward at time $t$. If two sides' preferences are known, the firm optimal stable matching is $\bar{u}_1 = \{[D_2, D_4], [S_5, S_1, S_3]\}$, $\bar{u}_2 = \{[D_3, D_1, D_5], [S_4, S_2]\}$ by the double matching algorithm. However, if firms' preferences are unknown, MMTS can learn these unknown preferences and attain the optimal stable matching while achieving a sublinear regret for each firm.

*MMTS Parameters.* We set priors $\alpha_{i,j}^{m,0} = \beta_{i,j}^{m,0} = 0.1, \forall i \in [N], \forall j \in [K_m], \forall m \in [M]$ to avoid too strong impact of the prior information. Each firm follows the MMTS algorithm to propose multiple types of workers. The update formula for each firm $p_i$ at time $t$ of the $m$-type worker $a_j^m$ is $\alpha_{i,j}^{m,t+1} = \alpha_{i,j}^{m,t} + 1$ if the worker $a_j^m$ is matched with the firm $p_i$, that is $a_j^m \in \mathbf{u}_t^m(p_i)$, and the received reward for firm $p_i$ is $y_{i,j}^m(t) = 1$; otherwise $\alpha_{i,j}^{m,t+1} = \alpha_{i,j}^{m,t}$; $\beta_{i,j}^{m,t+1} = \beta_{i,j}^{m,t} + 1$ if the worker $a_j^m$ is matched with the firm $p_i$ and the received reward for firm $p_i$ is $y_{i,j}^m(t) = 0$, otherwise $\beta_{i,j}^{m,t+1} = \beta_{i,j}^{m,t}$. For other unmatched pairs (firm, $m -$ type worker), the parameters retain.

**Results.** In Figure 2(b), we find that firm 1, 2 achieve a total *negative* sublinear regret and a total *positive* sublinear regret separately (solid lines). However, we find that due to the incorrect rankings provided by firms, firm 1 benefits from this non-optimal matching result in terms of *negative* sublinear regret specifically for matching with type 1 workers (blue dashed line). More discussion about the negative regret phenomenon is available in Appendix I.1.

**Example 2.** We enlarge the market by expanding the DS market, particularly wanting to explore interactions between two types of workers. $N = 2$ firms, $M = 2$ types, $K_1 = 20$ (DS) and $K_2 = 6$ (SDE). The DS quota for two firms is $q_1^1 = q_2^1 = 1$ and the SDE quota for two firms is $q_1^2 = q_2^2 = 3$, and the total quota is $Q_i = 6$ for both firms. Preferences from firms to workers and workers to firms are randomly generated. Therefore, the matching result for each firm should consist of three workers for each type, and type II workers will be fully allocated in the first match, and the rest workers are all type II workers. All MMTS initial parameters are the same as in Example 1.

**Results.** In Figure 2(c), we show when excessive type II workers exist, and type I workers are just right. Both firms can achieve positive sublinear regret. We find that since type II worker $K_2 = q_1^2 + q_2^2 = 6$, which means in the first match stage, those type II workers are fully allocated into two firms. Thus, in the second match stage, the left quota would be all allocated to the type I workers

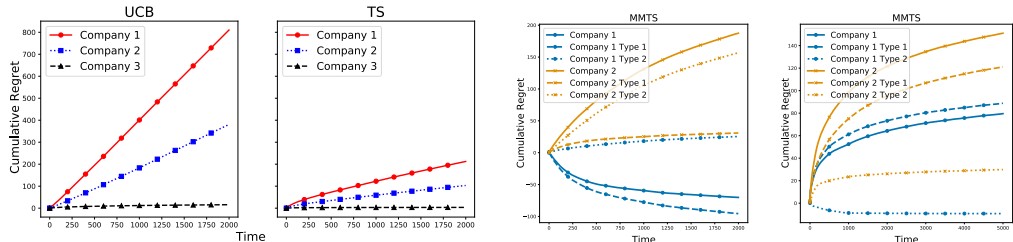

Figure 2: Left: A comparison of centralized UCB and TS. Right: firms and their sub-types regret for Example 1 and, firms and their sub-types regret for Example 2.

for two firms. Two dotted lines represent type II regret suffered by two firms. Both firms can quickly find the type II optimal matching since finding the optimal type II match just needs the first stage of the match. However, the type I workers' matching takes a longer time to find the optimal matching (take two stages), represented by dashed lines, and both are positive sublinear regret. Therefore, these two types of matching are fully independent, which is different from Example 1.

## 6    RELATED WORKS

We review multiple works in the literature, including matching while learning, multi-agent systems, assortment optimization, and matching markets. More can be found in Appendix J.

**Matching while Learning.** Liu et al. (2020) considers the multi-agent multi-armed competing problem in the centralized platform with explore-then-commit (ETC) and upper confidence bound (UCB) style algorithms where preferences from agents to arms are unknown and need to be learned through streaming interactive data. Jagadeesan et al. (2021) considers the two-sided matching problem where preferences from both sides are defined through dynamic utilities rather than fixed preferences and provide regret upper bounds over different contexts settings, and Min et al. (2022) apply it to the Markov matching market. Cen and Shah (2022) show that if there is transfer between agents, then the three desiderata (stability, low regret, and fairness) can be simultaneously achieved. Li et al. (2022) discuss the two-sided matching problem when the arm side has dynamic contextual information and preference is fixed from the arm side and propose a centralized contextual ETC algorithm to obtain the near-optimal regret bound. Besides, there are a plethora of works discussing the two-sided matching problem in the decentralized markets (Liu et al., 2021; Basu et al., 2021; Sankararaman et al., 2021; Dai and Jordan, 2021a;b; Dai et al., 2022). In particular, Dai and Jordan (2021b) study the college admission problem and provides an optimal strategy for agents, and shows its incentive-compatible property. Moreover, Jagadeesan et al. (2022) explores the phenomenon of the two-sided matching problem with two competing markets.

## 7    CONCLUSION AND FUTURE WORK

In this paper, we proposed a new algorithm, MMTS to solve the CMCP. MMTS builds on the strengths of TS for exploration and employs a *double matching* method to find a stable solution. Through theoretical analysis, we show the effectiveness of the algorithm in achieving stability at every matching step, achieving a sublinear Bayesian regret over time, and exhibiting the IC property.

There are several directions for future research. One is to investigate more efficient exploration strategies to reduce the time required to learn the agents' unknown preferences. Another is to examine scenarios where agents have indifferent preferences, and explore the optimal strategy for breaking ties. Additionally, it is of interest to incorporate real-world constraints such as budget or physical locations into the matching process, which could be studied using techniques from constrained optimization. Moreover, it is interesting to incorporate side information, such as agents' background information, into the matching process. This can be approached using techniques from recommendation systems or other machine learning algorithms that incorporate side information. Finally, it would be interesting to extend the algorithm to handle time-varying matching markets where preferences and the number of agents may change over time.

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
