This supplement is organized as follows. In Section A, we discuss the feasibility and its corresponding assumption of the stable matching. In Section B, we show the computational complexity of MMTS. In Section C, we exhibit why the centralized UCB suffers insufficient exploration. In Section D, we provide the Hoeffding concentration lemma. In Section E, we provide the stability property of MMTS. In Section F, we give the detailed proof of the regret upper bound of MMTS and decompose its proof into three parts, regret decomposition (F.1), bound for confidence width (F.2), and bad events' probabilities' upper bound (F.3). In Section G.1, we prove MMTS's strategy-proof property. Besides, as a reference, we append the DA with type and without type algorithms in Section H. In Section I, we provide details of experiments and the explanation of the negative regret, and also demonstrate the robustness of MMTS in large markets. Finally, in Section J, we provided additional related works.

## A    FEASIBILITY OF THE STABLE MATCHING

The feasibility solution is an interesting and well-discussed problem in the stable matching problem.

**Assumption of the feasibility:** In the finite market, it is the marginal preference assumption for the feasibility. But for the large market, it requires more assumptions such as the substitutability and indifferences, etc,. The difference between the infinite and finite (Azevedo and Hatfield, 2018; Greinecker and Kah, 2021) lies in matching problem and the techniques they use. In the infinite market, we assume that there is an uncountable number of agents on both sides of the market. This essentially means that the number of agents is so large that it can be treated as continuous, and you can't assign a specific numerical value to it. An example of an infinite market could be the matching of agents is extremely large and cannot be practically counted. In the finite market, the number of agents on both sides is limited and countable. You can assign a specific numerical value to the number of agents. An example could be the matching of agents where there is a definite small number of agents. However, such an exploration in the infinite market is beyond the scope of our current study.

In our case, if the complementary preference can be marginalized (or referred as the responsive preference (Roth, 1985), $(a_1, b_1) > (a_1, b_2)$ as long as $b_1 > b_2$, verse visa for $(a_1, b_1) > (a_2, b_1)$ as long as $a_1 > a_2$, which is at the top of Figure 1), then based on our proposed double matching algorithm and Theory 1, it exists such a stable matching solution. However, as discussed in the related works in Appendix J, if there exists couples in the preference list, which could potentially lead to an empty set of stable matchings.

Che et al. (2019) discussed that if there exists couples in the preference list in a infinite market (large) with a continuum of workers, provided that each firm's choice is convex and changes continuously as the set of available workers changes. They proved the existence and structure of stable matchings under preferences exhibiting substitutability and indifferences in a large market.

The difference between our result and (Che et al., 2019)'s result is in two ways: (1) we consider the finite market and they consider the infinite market. (2) we consider one side's preferences are unknown and (Che et al., 2019)'s both sides preferences are known. (3) Che et al. (2019) proved the existence of stable matching in the infinite market and no algorithm provided. However, in our paper, we provide the double matching algorithm to find it effectively.

## B    COMPLEXITY

Based on (Gale and Shapley, 1962; Knuth, 1997), the stable marriage problem's DA algorithm's worst total proposal number is $N^2 - 2N + 2 = \mathcal{O}(N^2)$ when the number of participants on both sides is equal ($N = K$). The computational complexity of the college admission matching problem with quota consideration is also $\mathcal{O}(NK)$. MMTS algorithm consists of two steps of matching. The computational complexity of the first step matching is $\mathcal{O}(\sum_{m=1}^{M} NK_m)$ if we virtually consider each type's matching process is organized in parallel. The second step's computation cost is also $\mathcal{O}(\sum_{m=1}^{M} NK_m)$. That is, in the first match, if all firms are matched with their best workers, this step meets the lower bound quota constraints. Then the second match will be reduced to the standard college admission problem without type consideration and the computational complexity

is $\mathcal{O}(N \sum_{m=1}^{M} K_m)$. So the total computational complexity is still $\mathcal{O}(\sum_{m=1}^{M} N K_m)$, which is polynomial in the of firm ($N$) and the number of workers $\sum_{m=1}^{M} K_m$ in the market.

## C  INCAPABLE EXPLORATION

In this section, we show why the TS strategy has an advantage over the UCB style method in estimating the ranks of workers. We even find that centralized UCB does achieve linear firm-optimal stable regret in some cases. In the following example (Example 6 from (Liu et al., 2020)), we show the firm achieves linear optimal stable regret if follow the UCB algorithm.[3]

Let $\mathcal{N} = \{p_1, p_2, p_3\}$, $\mathcal{K}_m = \{a_1, a_2, a_3\}$, and $M = 1$, with true preferences given below:

$$
\begin{aligned}
p_1 &: a_1 \succ a_2 \succ a_3 & \qquad a_1 &: p_2 \succ p_3 \succ p_1 \\
p_2 &: a_2 \succ a_1 \succ a_3 & \qquad a_2 &: p_1 \succ p_2 \succ p_3 \\
p_3 &: a_3 \succ a_1 \succ a_2 & \qquad a_3 &: p_3 \succ p_1 \succ p_2
\end{aligned}
$$

The firm optimal stable matching is $(p_1, a_1), (p_2, a_2), (p_3, a_3)$. However, due to incorrect ranking from firm $p_3$, $a_1 \succ a_3 \succ a_2$, and the output stable matching is $(p_1, a_2), (p_2, a_1), (p_3, a_3)$ based on the DA algorithm. In this case, $p_3$ will never have a chance to correct its mistake because $p_3$ will never be matched with $a_1$ again and cause the upper confidence bound for $a_1$ will never shrink and result in this rank $a_1 \succ a_3$. Thus, it causes that $p_1$ and $p_2$ suffer linear regret.

However, the TS is capable of avoiding this situation. By the property of sampling showed in Algorithm 2, firm $p_1$'s initial prior over worker $a_1$ is a uniform random variable, and thus $r_3(t) > r_1(t)$ (if we omit $a_2$) with probability $\hat{\mu}_3 \approx \mu_3$, rather than *zero*! This differs from the UCB style method, which cannot update $a_1$'s upper bound due to lacking exploration over $a_1$. The benefit of TS is that it can occasionally explore different ranking patterns, especially when there exists such a previous example.

In Figure 2(a), we show a quick comparison of centralized UCB (Liu et al., 2020) in the settings shown above and MMTS when $M = 1, Q = 1, N = 3, K = 3$. The UCB method occurs a linear regret in firm 1 and firm 2 and achieves a low matching rate $(0.031)$[4]. However, the TS method suffers a sublinear regret in firm 1 and firm 2 and achieves a high matching rate $(0.741)$. All results are averaged over 100 trials. See Section C.1 for the experimental details.

### C.1  SECTION 3.2 EXAMPLE - INSUFFICIENT EXPLORATION

We set the true matching reward for three firms to $(0.8, 0.4, 0.2), (0.5, 0.7, 0.2), (0.6, 0.3, 0.65)$. All preferences from companies over workers can be derived from the true matching reward. As we can view, company $p_3$ has a similar preference over $a_1$ (0.6) and $a_3$ (0.65). Thus, the small difference can lead the incapable exploration as described in Section 3.2 by the UCB algorithm.

## D  HOEFFDING LEMMA

**Lemma D.1.** *For any $\delta > 0$, with probability $1 - \delta$, the confidence width for a $m - type$ worker $a_j^m \in \mathcal{A}_{i,t}^m$ at time $t$ is upper bounded by*

$$
w_{i, \mathcal{F}_{i,t}^m}^m (a_j^m) \leq \min \left( 2 \sqrt{\frac{\log(\frac{2}{\delta})}{n_{i,j}^m(t)}}, 1 \right) \tag{C.1}
$$

*where $n_{i,j}^m(t)$ is the number of times that the pair $(p_i, a_j^m)$ has been matched at the start of round $t$.*

---

[3]Here we only consider one type of worker, and the firm's quota is one.

[4]We count 1 if the matching at time $t$ is fully equal to the optimal match when two sides' preferences are known. Then we take an average over the time horizon $T$.

*Proof.* Let $\hat{\mu}_{i,j,t}^{m,LS} = \frac{\sum_{s=1}^{t} \mathbf{1}(a_j^m \in \mathcal{A}_{i,s}^m) y_{i,j}^m(s)}{n_{i,j}^m(t)}$ denote the empirical mean reward from matching firm $p_i$ and $m-$ type worker $a_j^m$ up to time $t$. Define upper and lower confidence bounds as follows:

$$U_{i,t}^m(a_j^m) = \min\left\{\hat{\mu}_{i,j,t}^{m,LS} + \sqrt{\frac{\log(\frac{2}{\delta})}{n_{i,j}^m(t)}}, 1\right\}, L_{i,t}^m(a_j^m) = \max\left\{\hat{\mu}_{i,j,t}^{m,LS} - \sqrt{\frac{\log(\frac{2}{\delta})}{n_{i,j}^m(t)}}, 0\right\}. \quad \text{(C.2)}$$

The the confidence width is upper bounded by $\min\left(2\sqrt{\frac{\log(\frac{2}{\delta})}{n_{i,j}^m(t)}}, 1\right)$. $\qquad\square$

## E    PROOF OF THE STABILITY OF MMTS

*Proof.* We shall prove existence by giving an iterative procedure to find a stable matching.

**Part I**    To start, in the *first match* loop, based on the double matching procedure, we can discuss $M$ types of matching in parallel. So we will only discuss the path for seeking the type-$m$ company-worker stable matching.

Suppose firm $p_i$ has $q_i^m$ quota for $m$-type workers. We replace each firm $p_i$ by $q_i^m$ copies of $p_i$ denoted by $\{p_{i,1}, p_{i,2}, ..., p_{i,q_i^m}\}$. Each of these $p_{i,h}$ has preferences identical with those of $p_i$ but with a quota of 1. Further, each $m$-type worker who has $p_i$ on his/her preference list now replace $p_i$ by the set $\{p_{i,1}, p_{i,2}, ..., p_{i,q_i^m}\}$ in that order of preference. It is now easy to verify that the stable matchings for the firm $m$-type worker matching problem are in natural one-to-one correspondence with the stable matchings of this modified version problem. Then in the following, we only need to prove that stable matching exists in this transformed problem where each firm has quota 1, which is the standard stable marriage problem (Gale and Shapley, 1962). The existence of stable matching has been given in (Gale and Shapley, 1962). Here we reiterate it to help us to find the stable matching in the *second match*.

Let each firm propose to his favorite $m$-type worker. Each worker who receives more than one offer rejects all but her favorite from among those who have proposed to her. However, the worker does not fully accept the firm, but keeps the firm on a string to allow for the possibility that some better firm come along later.

Now we are in the second stage. Those firms who were rejected in the first stage propose to their second choices. Each $m$-type worker receiving offers chooses her favorite from the group of new firms and the firm on her string, if any. The worker rejects all the rest and again keeps the favorite in suspense. We proceed in the same manner. Those firms who are rejected at the second stage propose to their next choices, and the $m$-type workers again reject all but the best offer they have had so far.

Eventually, every $m$-type worker will have rejected a proposal, for as long as any worker has not been proposed to there will be rejections and new offers[5], but since no firm can propose the same $m$-type worker more than once, every worker is sure to get a proposal in due time. As soon as the last worker gets her offer, the "recruiting" is declared over, and each $m$-type worker is now required to accept the firm on her string.

We asset that this set of matching is stable. Suppose firm $p_i$ and $m$-type worker $a_j$ are not matched to each other but firm $p_i$ prefers $a_j$ to his current matching $m$-type worker $a_{j'}$. Then $p_i$ must have proposed to $a_j$ at some stage (since the proposal is ordered by the preference list) and subsequently been rejected in favor of some firm $p_{i'}$ that $a_j$ liked better. It is clear that $a_j$ must prefer her current matching firm $p_{i'}$ and there is no instability/blocking pair.

Thus, each $m$-type firm-worker matching established on the first match is stable. Then each firm $p_i$'s matching object in the first match with quota $q_i^m$ can be recovered as grouping all matching objects of firm $\{p_{i,h}\}_{h=1}^{q_i^m}$.

**Part II**    To start the second match, we first check the left quota $\tilde{Q}_i$ for each firm. If the left quota is zero for firm $p_i$, then firm $p_i$ and its matching workers will quit the matching market and get its stable matching object. Otherwise, the left firm will continue to participate in the second match.

---

[5]Here we assume the number of firms is less than or equal to the number workers, and those workers unmatched finally will be matched to themselves and assume their matching object is on the firm side.

In the second match, preferences from firms to workers are un-categorized. Based on line 19 in Algorithm 3, all types of workers will be ranked to fill the left quota. Thus, it reduces to the problem in part I, and the result matching in the second match is also stable. What is left to prove is that the overall double matching algorithm can provide stable matching. In the second match, each firm proposes to workers in his left concatenate ordered preference list, and all previous workers not in the second match preference list have already been matched or rejected. So it cannot form a blocking pair between the firm $p_i$ with leftover workers. $\qquad\square$

## F  MMTS REGRET UPPER BOUND

### F.1  REGRET DECOMPOSITION

In this part, we provide the road map of the regret decomposition and key steps to prove Theorem 4.2. First, we define the history for firm $p_i$ up to time $t$ of type $m$ as $H_{i,t}^m :=$ $\{\mathcal{A}_{i,1}^m, \mathbf{y}_{i,\mathcal{A}_{i,1}^m}^m(1), \mathcal{A}_{i,2}^m, \mathbf{y}_{i,\mathcal{A}_{i,2}^m}^m(2), ..., \mathcal{A}_{i,t-1}^m, \mathbf{y}_{i,\mathcal{A}_{i,t-1}^m}^m(t-1)\}$, composed by actions (matched workers) and rewards, where $\mathcal{A}_{i,t}^m := \mathbf{u}_t^m(p_i)$ is a set of workers (based on quota requirement $q_i^m$ and $Q_i$) belong to $m$-type which is matched with firm $p_i$ at time $t$, $\mathbf{y}_{i,\mathcal{A}_{i,t-1}^m}^m(t-1)$ are realized rewards when firm $p_i$ matched with $m-$ type workers $\mathcal{A}_{i,t}^m$. Define $\widetilde{H}_{i,t} := \{H_{i,t}^1, H_{i,t}^2, ..., H_{i,t}^M\}$ as the aggregated interaction history between firm $p_i$ and all types of workers up to time $t$.

Next, we define the *good event* for firm $p_i$ when matching with $m-$ type worker at time $t$ and the true mean matching reward falls in the uncertainty set as $E_{i,t}^m = \{\boldsymbol{\mu}_{i,\mathcal{A}_{i,t}^m}^m \in \mathcal{F}_{i,t}^m\}$, where $\boldsymbol{\mu}_{i,\mathcal{A}_{i,t}^m}^m$ is the true mean reward vector of actually pulled arms (matched with $m-$ type workers) at time $t$ for firm $p_i$, and $\mathcal{F}_{i,t}^m$ is the uncertainty set for $m-$ type worker at time $t$ for firm $p_i$. Similarly, the good event for firm $p_i$ when matching with all types of workers at time $t$ is $E_{i,t} = \bigcap_{m=1}^M E_{i,t}^m$, over all firms $E_t = \bigcap_{i=1}^N E_{i,t}$. And the corresponding *bad event* is defined as $\overline{E}_{i,t}^m, \overline{E}_{i,t}, \overline{E}_t$ respectively. That represents the true mean vector/tensor reward of the pulled arms is not in the uncertainty set.

**Lemma F.1.** *Fix any sequence $\{\widetilde{\mathcal{F}}_{i,t} : i \in [N], t \in \mathbb{N}\}$, where $\widetilde{\mathcal{F}}_{i,t} \subset \mathcal{F}$ is measurable with respect to $\sigma(\widetilde{H}_{i,t})$. Then for any $T \in \mathbb{N}$, with probability 1,*

$$\mathfrak{R}(T) \leq \mathbb{E}\sum_{t=1}^T \left[ \sum_{i=1}^N \sum_{m=1}^M \widetilde{W}_{i,\mathcal{F}_{i,t}^m}^m(\mathcal{A}_{i,t}^m) + C\mathbf{1}(\overline{E}_t) \right] \tag{C.3}$$

*where $\widetilde{W}_{i,\widetilde{\mathcal{F}}_{i,t}^m}^m(\cdot) = \sum_{a_j^m \in \mathcal{A}_{i,t}^m} w_{i,\widetilde{\mathcal{F}}_{i,t}^m}^m(a_j^m)$ represents the sum of the element-wise value of uncertainty width at $m-$ type worker $a_j^m$. The uncertainty width $w_{i,\widetilde{\mathcal{F}}_{i,t}^m}^m(a_j^m) = \sup\limits_{\overline{\mu}_i^m, \underline{\mu}_i^m \in \widetilde{\mathcal{F}}_{i,t}^m}(\overline{\mu}_i^m(a_j^m) - \underline{\mu}_i^m(a_j^m))$ is a worst-case measure of the uncertain about the mean reward of $m-$ type worker $a_j^m$. Here $C$ is a constant less than 1.*

*Proof.* The key step of regret decomposition is to split the instantaneous regret by firms, types, and quotas. Then we categorize regret by the happening of good events and bad events. The good events' regret is measured by the uncertainty width, and the bad events' regret is measured by the probability of happening it.

To reduce notation, define element-wise upper and lower bounds $U_{i,t}^m(a) = \sup\{\mu_i^m(a) : \mu_i^m \in \mathcal{F}_{i,t}^m, a \in \mathcal{K}_m\}$ and $L_{i,t}^m(a) = \inf\{\mu_i^m(a) : \mu_i^m \in \mathcal{F}_{i,t}^m, a \in \mathcal{K}_m\}$, where $\mu_i^m$ is the mean reward function $\mu_i^m \in \mathcal{F}_i^m : \mathbb{R} \mapsto \mathbb{R}, \forall i \in [N], \forall m \in [M]$. Whenever $\mu_{i,\widetilde{\mathcal{A}}_i^m}^m \in \mathcal{F}_{i,t}^m$, the bounds $L_{i,t}^m(a) \leq \mu_{i,\widetilde{\mathcal{A}}_i^m}^m(a) \leq U_{i,t}^m(a)$ hold for all types of workers. Here we define $\mathcal{A}_{i,t}^m = \mathbf{u}_i^m(t)$ as the matched $m-$ type workers for firm $p_i$ at time $t$ and $\mathcal{A}_{i,t}^{m,*} = \overline{\mathbf{u}}_i^m(t)$ as the firm $p_i$'s optimal stable matching result of $m-$ type workers at time $t$. Since the firm-optimal stable matching result is fixed, given both sides' preferences, we can omit time $t$ here. The firm-optimal stable matching result set is also denoted as $\mathcal{A}_i^{m,*} = \mathcal{A}_{i,t}^{m,*}$.

As for type-$m$ workers' matching for the firm $p_i$ at time $t$, the instantaneous regret with a given instance $\theta$ can be implied as follows, here for simplicity, we omit the instance conditional notation

$$
\begin{aligned}
\mathcal{I}_{i,t}^m = \mu_i^m(\mathcal{A}_i^{m,*}) - \mu_i^m(\mathcal{A}_{i,t}^m) &\leq \sum_{a \in \mathcal{A}_i^{m,*}} U_{i,t}^m(a) - \sum_{a \in \mathcal{A}_{i,t}^m} L_{i,t}^m(a) + C\mathbf{1}(\boldsymbol{\mu}_{i,\widetilde{\mathcal{A}}_i}^m \notin \mathcal{F}_{i,t}^m) \\
&= \widetilde{U}_{i,t}^m(\mathcal{A}_i^{m,*}) - \widetilde{L}_{i,t}^m(\mathcal{A}_{i,t}^m) + C\mathbf{1}(\boldsymbol{\mu}_{i,\widetilde{\mathcal{A}}_i}^m \notin \mathcal{F}_{i,t}^m) \\
&= \widetilde{W}_{i,\mathcal{F}_{i,t}^m}(\mathcal{A}_{i,t}^m) + [\widetilde{U}_{i,t}^m(\mathcal{A}_i^{m,*}) - \widetilde{U}_{i,t}^m(\mathcal{A}_{i,t}^m)] + C\mathbf{1}(\boldsymbol{\mu}_{i,\widetilde{\mathcal{A}}_i}^m \notin \mathcal{F}_{i,t}^m),
\end{aligned}
\tag{C.4}
$$

where $C \leq 1$ is a constant, and we let $\widetilde{U}_{i,t}^m(\cdot) = \sum_a U_{i,t}^m(a)$ and $\widetilde{W}_{i,\mathcal{F}_{i,t}^m}(\cdot) = \sum_a w_{i,\mathcal{F}_t}^m(a)$ represent the sum of the element-wise value of $U_{i,t}^m(\cdot), w_{i,\mathcal{F}_{i,t}}^m(\cdot)$, respectively. Define the good event for firm $p_i$, matching with $m-$ type worker at time $t$ is $E_{i,t}^m = \{\boldsymbol{\mu}_{i,\widetilde{\mathcal{A}}_i}^m \in \mathcal{F}_{i,t}^m\}$, over all types $E_{i,t} = \bigcap_{m=1}^M E_{i,t}^m$, over all firms $E_t = \bigcap_{i=1}^N E_{i,t}$. And the corresponding bad event is defined as $\overline{E}_{i,t}^m, \overline{E}_{i,t}, \overline{E}_t$ respectively.

Now consider Eq. (C.3), summing over the previous equation over time $t$, firms $p_i$, and workers' type $m$, we get

$$
\begin{aligned}
\mathfrak{R}(T) &\leq \mathbb{E}\sum_{i=1}^N \sum_{t=1}^T \sum_{m=1}^M [\widetilde{W}_{i,\mathcal{F}_{i,t}^m}(\mathcal{A}_{i,t}^m) + C\mathbf{1}(\overline{E}_t)] + \sum_{i=1}^N \mathbb{E}M_{i,T} \\
&= \mathbb{E}\sum_{t=1}^T [C\mathbf{1}(\overline{E}_t) + \sum_{i=1}^N \sum_{m=1}^M \widetilde{W}_{i,\mathcal{F}_{i,t}^m}(\mathcal{A}_{i,t}^m)] + \sum_{i=1}^N \mathbb{E}M_{i,T}
\end{aligned}
\tag{C.5}
$$

where $M_{i,T} = \sum_{t=1}^T \sum_{m=1}^M [\widetilde{U}_{i,t}^m(\mathcal{A}_i^{m,*}) - \widetilde{U}_{i,t}^m(\mathcal{A}_i^m)]$. Now by the definition of TS, $\mathbb{P}_m(\mathcal{A}_{i,t}^m \in \cdot | H_{i,t}^m) = \mathbb{P}_m(\mathcal{A}_i^{m,*} \in \cdot | H_{i,t}^m)$ for all types, where $\mathbb{P}_m(\cdot | H_{i,t}^m)$ represents this probability is conditional on history $H_{i,t}^m$ and the selected action (worker) belongs in $m$-type workers for firm $p_i$. That is $\mathcal{A}_{i,t}^m$ and $\mathcal{A}_i^{m,*}$ within type-$m$ is identically distributed under the posterior. Besides, since the confidence set $\mathcal{F}_{i,t}^m$ is $\sigma(H_{i,t}^m)$-measurable, so is the induced upper confidence bound $U_{i,t}^m(\cdot)$. This implies $\mathbb{E}_m[U_{i,t}^m(\mathcal{A}_{i,t}^m)|H_t^m] = \mathbb{E}_m[U_{i,t}^m(\mathcal{A}_i^{m,*})|H_t^m]$, and there for $\mathbb{E}[M_{i,T}] = 0$ and $\sum_{i=1}^N \mathbb{E}M_{i,T} = 0$. Then we can obtain the desired result. $\qquad\square$

## F.2 Uncertainty Widths

In this part, we provide the upper bound of the accumulated uncertainty widths over all types of workers and all firms, which is the first part in Eq. (C.3).

**Lemma F.2.** *If $(\beta_{i,j,t}^m \geq 0 | t \in \mathbb{N})$ is a non-decreasing sequence and $\mathcal{F}_{i,j,t}^m := \{\mu_{i,j}^m \in \mathcal{F}_{i,j}^m : \left\| \mu_{i,j}^m - \hat{\mu}_{i,j,t}^{m,LS} \right\|_1 \leq \sqrt{\beta_{i,j,t}^m}\}$, then with probability 1,*

$$
\sum_{t=1}^T \sum_{i=1}^N \sum_{m=1}^M \widetilde{W}_{i,\mathcal{F}_{i,t}^m}^m(\mathcal{A}_{i,t}^m) \leq 8Q\log(QT)\sqrt{K_{\max}T}.
$$

The proof of this lemma builds upon Lemma F.3, which establishes the number of instances where the widths of uncertainty sets for a chosen set of $m-$ type workers $\mathcal{A}_{i,t}^m$ greater than $\epsilon$. We show that this number is determined by the *Eluder dimension* (Russo and Van Roy, 2014).

*Proof.* By Lemma F.1, the instantaneous regret $\mathcal{I}_t$ over all firms and all types, can be decomposed by types and by firms and shown as

$$
\begin{aligned}
\mathcal{I}_t = \sum_{m=1}^{M} \mathcal{I}_t^m = \sum_{i=1}^{N} \sum_{m=1}^{M} \mathcal{I}_{i,t}^m & \\
\leq \sum_{i=1}^{N} \sum_{m=1}^{M} \widetilde{W}_{i,\mathcal{F}_{i,t}^m}(\mathcal{A}_{i,t}^m), \quad & \text{if } E_t \text{ holds.} \\
\leq 2 \sum_{i\in[N], m\in[M], a_j^m \in \mathcal{K}_m} \sqrt{\frac{\log(\sum_{i=1}^{N} Q_i T)}{n_{i,j}^m(t)}}, \quad & \text{with prob } 1-\delta
\end{aligned}
\tag{C.6}
$$

where the first inequality is based on Lemma F.1 and if $E_t$ holds for $t \in \mathbb{N}, m \in M, i \in [N]$, $n_{i,j}^m(t)$ is the number of times that the pair $(p_i, a_j^m)$ has been matched at the start of round $t$. The second inequality is constructed from a union concentration inequality based on Lemma D.1, and we set $\delta = 2/\sum_{i=1} Q_i T$. We denote $z_{i,j}^m(t) = \frac{1}{\sqrt{n_{i,j}^m(t)}}$ as the size of the scaled confidence set (without the log factor) for the pair $(p_i, a_j^m)$ at the time $t$.

At each time step $t$, let's consider the list consisting of $z_{i,j}^m(t)$ and reorder the overall list consisting of concatenating all those scaled confidence sets over all rounds and all types in decreasing order. Then we obtain a list $\tilde{z}_1 \geq \tilde{z}_2 \geq ..., \geq \tilde{z}_L$, where $L = \sum_{t=1}^{T} \sum_{i=1}^{N} Q_i = T \sum_{i=1}^{N} Q_i$. We reorganize the Eq. (C.6) to get

$$
\sum_{t=1}^{T} \mathcal{I}_t \leq \sum_{t=1}^{T} \sum_{m=1}^{M} \sum_{i=1}^{N} \widetilde{W}_{i,\mathcal{F}_{i,t}^m}(\mathcal{A}_{i,t}^m) \leq 2\log(\sum_{i=1}^{N} Q_i T) \sum_{l=1}^{L} \tilde{z}_l.
\tag{C.7}
$$

By Lemma F.3, the number of rounds that a pair of a firm and any $m-$ type worker can have it confidence set have size at least $\tilde{z}_l$ is upper bounded by $(1 + \frac{4}{\tilde{z}_l^2}) K_m$ when we set $\epsilon = \tilde{z}_l$ and know $\beta_{i,j,t}^m \leq 1$. Thus, the total number of times that any confidence set can have size at least $\tilde{z}_l$ is upper bounded by $(1 + \frac{4}{\tilde{z}_l^2}) \sum_{i=1}^{N} \sum_{m=1}^{M} |\mathcal{A}_{i,t}^m| K_m$. To determine the minimum condition for $\tilde{z}_l$, which is equivalent to determine the maximum of $l$, we have $l \leq (1 + \frac{4}{\tilde{z}_l^2}) \sum_{i=1}^{N} \sum_{m=1}^{M} |\mathcal{A}_{i,t}^m| K_m$. So we claim that

$$
\tilde{z}_l \leq \min\left(1, \frac{2}{\sqrt{\frac{l}{\sum_{i=1}^{N} \sum_{m=1}^{M} |\mathcal{A}_{i,t}^m| K_m} - 1}}\right) \leq \min\left(1, \frac{2}{\sqrt{\frac{l}{\sum_{i=1}^{N} Q_i K_{\max}} - 1}}\right),
\tag{C.8}
$$

where the second inequality above is by $\sum_{i=1}^{N} \sum_{m=1}^{M} |\mathcal{A}_{i,t}^m| K_m \leq K_{\max} \sum_{i=1}^{N} \sum_{m=1}^{M} |\mathcal{A}_{i,t}^m| \leq K_{\max} \sum_{i=1}^{N} Q_i = Q K_{\max}$ and $K_{\max} = \max\{K_1, ..., K_M\}, Q = \sum_{i=1}^{N} Q_i$. Putting all these together, we have

$$
\begin{aligned}
2\log(\sum_{i=1}^{N} Q_i T) \sum_{l=1}^{L} \tilde{z}_l &\leq 2\log(QT) \sum_{l=1}^{L} \min(1, \frac{2}{\sqrt{\frac{l}{Q K_{\max}} - 1}}) \\
&= 4\log(QT) \sum_{l=1}^{QT} \frac{1}{\sqrt{\frac{l}{Q K_{\max}} - 1}} \\
&\leq 8\log(QT)\sqrt{Q K_{\max}}\sqrt{QT}
\end{aligned}
\tag{C.9}
$$

where the last inequality is by intergral inequality

$$
\sum_{l=1}^{QT} \frac{1}{\sqrt{\frac{l}{Q K_{\max}} - 1}} \leq \sqrt{Q K_{\max}} \sum_{l=1}^{QT} \frac{1}{\sqrt{l}} \leq \sqrt{Q K_{\max}} \int_{x=0}^{QT} \frac{1}{\sqrt{x}} dx = 2\sqrt{Q K_{\max}}\sqrt{QT}.
$$

Based on Eq. (C.7) and the above result, we can get the regret

$$
\sum_{t=1}^{T} \mathcal{I}_t \leq 8Q\log(QT)\sqrt{K_{\max}T},
\tag{C.10}
$$

if $E_t$ holds. $\qquad \square$

**Lemma F.3.** *If $(\beta_{i,j,t}^m \geq 0 | t \in \mathbb{N})$ is a nondecreasing sequence for $i \in [N], a_j^m \in \mathcal{K}_m, m \in [M]$ and $\mathcal{F}_{i,j,t}^m := \{\mu_{i,j}^m \in \mathcal{F}_{i,j}^m : \left\| \mu_{i,j}^m - \hat{\mu}_{i,j,t}^{m,LS} \right\|_1 \leq \sqrt{\beta_{i,j,t}^m}\}$, for all $T \in \mathbb{N}$ and $\epsilon > 0$, then*

$$\sum_{t=1}^{T} \sum_{m=1}^{M} \sum_{a_j^m \in \mathcal{A}_{i,t}^m} \mathbf{1}\big(w_{i,\mathcal{F}_{i,t}^m}^m(a_j^m) > \epsilon\big) \leq \big(\frac{4\widetilde{\beta}_{i,T}}{\epsilon^2} + 1\big) \sum_{m=1}^{M} |\mathcal{A}_{i,t}^m| K_m.$$

*Here $\hat{\mu}_{i,j,t}^{m,LS} = \frac{\sum_{s=1}^{t} \mathbf{1}(a_j^m \in \mathcal{A}_{i,s}^m) y_{i,j}^m(s)}{n_{i,j}^m(t)}$ is the estimated average reward for $m-$ type worker $a_j^m$ from the view point of firm $p_i$ at time $t$, and $n_{i,j}^m(t)$ is the number of matched times up to time $t$ of firm $p_i$ with $m-$ type worker $a_j^m$. Besides, we define $\widetilde{\beta}_{i,T} = \max_{a_j^m \in \mathcal{K}_m, m \in [M]} \beta_{i,j,T}^m$ as the maximum uncertainty bound over all types of workers at time $T$ for firm $p_i$.*

The proof of this result is based on techniques from (Russo and Van Roy, 2013; 2014). This result demonstrates that the upper bound of the number of times the widths of uncertainty sets exceeds $\epsilon$ is dependent on the error $\mathcal{O}(\epsilon^{-2})$ and linearly proportional to the product of the number of $m-$ type worker and the type quota size $q_i^m$.

*Proof.* Based on the Proposition 3 from (Russo and Van Roy, 2013), we can use the *eluder dimension* $dim_E(\mathcal{F}_i^m, \epsilon)$ to bound the number of times the widths of confidence intervals for a selection of set of $m-$ type workers $\mathcal{A}_{i,t}^m$ greater than $\epsilon$.

$$\sum_{t=1}^{T} \sum_{m=1}^{M} \sum_{a_j^m \in \mathcal{A}_{i,t}^m} \mathbf{1}\left(w_{i,\mathcal{F}_{i,t}^m}^m(a_j^m) > \epsilon\right) \leq \sum_{m=1}^{M} \sum_{a_j^m \in \mathcal{A}_{i,t}^m} \left(\frac{4\beta_{i,j,T}^m}{\epsilon^2} + 1\right) dim_E(\mathcal{F}_i^m, \epsilon)$$

$$\leq \left(\frac{4 \max\limits_{a_j^m \in \mathcal{K}_m, m \in [M]} \beta_{i,j,T}^m}{\epsilon^2} + 1\right) \sum_{m=1}^{M} |\mathcal{A}_{i,t}^m| dim_E(\mathcal{F}_i^m, \epsilon),$$

(C.11)

where the eluder dimension of a multi-arm bandit problem is the number of arms, we get

$$\sum_{t=1}^{T} \sum_{m=1}^{M} \sum_{a_j^m \in \mathcal{A}_{i,t}^m} \mathbf{1}\left(w_{i,\mathcal{F}_t}^m(a_j^m) > \epsilon\right) \leq \left(\frac{4\widetilde{\beta}_{i,T}}{\epsilon^2} + 1\right) \sum_{m=1}^{M} |\mathcal{A}_{i,t}^m| K_m \leq \left(\frac{4\widetilde{\beta}_{i,T}}{\epsilon^2} + 1\right) Q_i K_{\max}$$

(C.12)

where $\widetilde{\beta}_{i,T} = \max\limits_{a_j^m \in \mathcal{K}_m, m \in [M]} \beta_{i,j,T}^m$. Besides, we know that $Q_i = \sum_{m=1}^{M} |\mathcal{A}_{i,t}^m|$ and define $K_{\max} = \max\limits_{m \in [M]} K_m$, so we can get the second inequality. $\square$

### F.3 BAD EVENT UPPER BOUND

In this part, we provide an upper bound of the second part of Eq. (C.3). The regret caused by the happening of the bad event at each time step is quantified by the following lemma.

**Lemma F.4.** *If $\mathcal{F}_{i,j,t}^m := \{\mu_{i,j}^m \in \mathcal{F}_{i,j}^m : \left\| \mu_{i,j}^m - \hat{\mu}_{i,j,t}^{m,LS} \right\|_1 \leq \sqrt{\beta_{i,j,t}^m}\}$ holds with probability $1 - \delta$, then the bad event $\overline{E}_t$ happening's probability is upper bounded by $\mathbb{E}\mathbf{1}(\overline{E}_t) \leq NK\delta$. In particular, if $\delta = 1/QT$, the accumulated bad events' probability is upper bounded by $\sum_{t=1}^{T} \mathbb{E}\mathbf{1}(\overline{E}_t) \leq NK/Q$.*

To bound the probability of bad events, we use a union bound to obtain the desired result. Specifically, if $Q_i = 1$, which means each firm has a total quota of 1 and only considers one type of worker, then $\sum_{t=1}^{T} \mathbb{E}\mathbf{1}(\overline{E}_t) \leq NK/(N \times 1) = K$. This shows that each firm needs to explore a single type of worker, and the worst total regret is less than $K$. If $Q_i = 1, M = 1$, which means all firms have the same recruiting requirements, the result reduces to the general competitive matching scenario, and the worst regret is the number of workers of type $K_M$ in the market.

*Proof.* If $E_t$ does not hold, the probability of the true matching reward is not in the confidence interval we constructed is upper bounded by

$$\mathbb{E}\mathbf{1}(\overline{E}_t) = \mathbb{P}(\overline{E}_t) = \mathbb{P}\Bigg(\Big(\bigcap_{i\in[N]}\bigcap_{m\in[M]}\bigcap_{a_j^m\in\mathcal{K}_m}\{\mu_{i,j}^m\in\mathcal{F}_{i,j,t}^m\}\Big)^c\Bigg)$$

$$= \mathbb{P}\Bigg(\bigcup_{i\in[N]}\bigcup_{a_j^m\in\mathcal{K}_m}\bigcup_{m\in[M]}\{\mu_{i,j}^m\notin\mathcal{F}_{i,j,t}^m\}\Bigg)$$

$$= \mathbb{P}\Bigg(\bigcup_{i\in[N]}\bigcup_{a_j^m\in\mathcal{K}_m}\bigcup_{m\in[M]}\Big\{\big\|\mu_{i,j}^m-\hat{\mu}_{i,j,t}^{m,LS}\big\|_{2,E_t}\geq\sqrt{\beta_{i,j,t}^m}\Big\}\Bigg) \qquad \text{(C.13)}$$

$$= \mathbb{P}\Bigg(\bigcup_{i\in[N]}\bigcup_{a_j^m\in\mathcal{K}_m}\bigcup_{m\in[M]}\Big\{\big\|\mu_{i,j}^m-\hat{\mu}_{i,j,t}^{m,LS}\big\|_1\geq\sqrt{\frac{\log(\frac{2}{\delta})}{n_{i,j}^m(t)}}\Big\}\Bigg)$$

$$\leq \sum_{i\in[N]}\sum_{a_j^m\in\mathcal{K}_m}\sum_{m\in[M]}\mathbb{P}\Bigg(\big\|\mu_{i,j}^m-\hat{\mu}_{i,j,t}^{m,LS}\big\|_1\geq\sqrt{\frac{\log(\frac{2}{\delta})}{n_{i,j}^m(t)}}\Bigg)$$

where the third equality is by De-Morgan's Law of sets. In the last inequality, we use the union bound to control the probability. Since each $\hat{\mu}_{i,j}^{m,LS}-\mu_{i,j}^m$ is a mean zero and $\frac{1}{2n_{i,j}^m}$-sub-Gaussian random variable, based on Lemma D.1, have $\mathbb{P}\big(\big\|\mu_{i,j}^m-\hat{\mu}_{i,j,t}^{m,LS}\big\|_1\geq\sqrt{\frac{\log(\frac{2}{\delta})}{n_{i,j}^m(t)}}\big)\leq\delta$. The overall bad event's probability's upper bound is

$$\mathbb{P}(\overline{E}_t)\leq NK\delta \qquad \text{(C.14)}$$

Based on our confidence width is less than 1, so $C=1,\forall i\in[N]$. The expected regret from this bad event is not in the confidence interval at most

$$NK\delta\cdot CT\leq NK\frac{1}{\sum_{i=1}^N Q_iT}T=\frac{NK}{Q} \qquad \text{(C.15)}$$

This part's regret is negligible compared with the regret from Lemma F.2. In particular, if there is only one type and each firm has only one position to be filled. Thus, $Q=N$, the bad event's upper bounded probability will shrink to $K$, the number of workers to be explored. $\qquad\square$

In this part, we provide the proof of MMTS's Bayesian regret upper bound.

## F.4    PROOF OF THEOREM 4.2

**Theorem F.1.** *When all firms follow the MMTS algorithm, the platform will incur the Bayesian total expected regret*

$$\mathfrak{R}(T)\leq 8\log(QT)\sqrt{QK_{\max}}\sqrt{QT}+NK/Q \qquad \text{(C.16)}$$

*where $K_{\max}=\max\{K_1,...,K_M\},K=\sum_{m=1}^M K_m$ .*

*Proof.* We decompose the Bayesian total firm-optimal stable regret for all firms by

$$\mathfrak{R}(T)=\mathbb{E}_{\theta\in\Theta}\Bigg[\sum_{i=1}^N R_i(T,\theta)\Bigg]=\mathbb{E}_{\theta\in\Theta}\Bigg[\sum_{i=1}^N\sum_{m=1}^M\sum_{t=1}^T\mu_{i,\overline{u}_i^m(t)}(t)-\sum_{i=1}^N\sum_{m=1}^M\sum_{t=1}^T\mu_{i,u_i^m}(t)|\theta\Bigg]$$

$$= \sum_{i=1}^N\sum_{t=1}^T\mathbb{E}_{\theta\in\Theta}\Bigg[\sum_{m=1}^M(\mu_{i,\overline{u}_i^m(t)}(t)-\mu_{i,u_i^m}(t))|\theta\Bigg]$$

$$= \mathbb{E}_{\theta\in\Theta}\Bigg[\sum_{t=1}^T\sum_{i=1}^N\sum_{m=1}^M\mathcal{I}_{i,t}^m|\theta\Bigg]$$

$$= \mathbb{E}_{\theta\in\Theta}\Bigg[\sum_{t=1}^T\mathcal{I}_t|\theta\Bigg]$$

$$\text{(C.17)}$$

where we define $\mathcal{I}_{i,t}^m = \mu_{i,\boldsymbol{\theta}}^m(\mathcal{A}_i^{m,*}) - \mu_{i,\boldsymbol{\theta}}^m(\mathcal{A}_{i,t}^m)$ and $\mathcal{I}_t = \sum_{i=1}^N \sum_{m=1}^M \mathcal{I}_{i,t}^m$. Here $\mathcal{A}_i^{m,*}$ is the optimal matched workers for firm $p_i$ of type $m$ and $\mathcal{A}_{i,t}^m$ is the actual matched workers for firm $p_i$ of type $m$ at time $t$ under the instance $\theta$.

Based Lemma F.1, $\mathfrak{R}(T)$ is upper bounded by $\mathbb{E} \sum_{t=1}^T \left[ C\mathbf{1}(\overline{E}_t) + \sum_{i=1}^N \sum_{m=1}^M \widetilde{W}_{i,\mathcal{F}_{i,t}^m}(\mathcal{A}_{i,t}^m) \right]$. The first term, the sum of the bad event probability $\mathbb{E} \sum_{t=1}^T C\mathbf{1}(\overline{E}_t) = C \sum_{t=1}^T \mathbb{P}(\overline{E}_t)$, which is upper bounded by $NK/Q$ based on Lemma F.4 and $C \le 1$. The second term, the sum of confidence widths is upper bounded by $8Q\log(QT)\sqrt{TK_{\max}}$ based on Lemma F.2. Thus the Bayesian total regret is upper bounded by $8Q\log(QT)\sqrt{TK_{\max}} + NK/Q$. □

## G  Incentive-Compatibility

In this section, we discuss the incentive-compatibility property of MMTS. That is, if one firm does not follow the MMTS when all other firms submit their MMTS preferences, that firm cannot benefit (matched with a better worker than his optimal stable matching worker) over a sublinear order. As we know, Dubins and Freedman (1981) discussed the *Machiavelli* firm could not benefit from incorrectly stating their true preference when there exists a unique stable matching. However, when one side's preferences are unknown and need to be learned through data, this result no longer holds. Thus, the maximum benefits that can be gained by the Machiavelli firm are under-explored in the setting of learning in matching. Liu et al. (2020) discussed the benefits that can be obtained by Machiavelli firm when other firms follow the centralized-UCB algorithm with the problem setting of one type of worker and quota equal one in the market.

We now show in CMCP, when all firms except one $p_i$ submit their MMTS-based preferences to the matching platform, the firm $p_i$ has an incentive also to submit preferences based on their sampling rankings in a *long horizon*, so long as the matching result do not have multiple stable solutions. Now we establish the following lemma, which is an upper bound of the expected number of pulls that a firm $p_i$ can match with a $m$-type worker that is better than their optimal $m$-type workers, regardless of what preferences they submit to the platform.

Let's use $\mathcal{H}_{i,l}^m$ to define the achievable *sub-matching* set of $\mathbf{u}^m$ when all firms follow the MMTS, which represents firm $p_i$ and $m-$ type worker $a_l^m$ is matched such that $a_l^m \in \mathbf{u}_i^m$. Let $\Upsilon_{\mathbf{u}^m}(T)$ be the number of times sub-matching $\mathbf{u}^m$ is played by time $t$. We also provide the blocking triplet in a matching definition as follows.

**Definition 3.** *(Blocking triplet) A blocking triplet $(p_i, a_k, a_{k'})$ for a matching $u$ is that there must exist a firm $p_i$ and worker $a_j$ that they both prefer to match with each other than their current match. That is, if $a_{k'} \in \mathbf{u}_i$, $\mu_{i,k'} < \mu_{i,k}$ and worker $a_k$ is either unmatched or $\pi_{k,i} < \pi_{k,\mathbf{u}^{-1}(k)}$.*

The following lemma presents the upper bound of the number of matching times of $p_i$ and $a_l^m$ by time $T$, where $a_l^m$ is a *super optimal $m-$ type worker* (preferred than all stable optimal $m-$ type workers under true preferences), when all firms follow the MMTS.

**Lemma G.1.** *Let $\Upsilon_{i,l}^m(T)$ be the number of times a firm $p_i$ matched with a $m$-type worker such that the mean reward of $a_l^m$ for firm $p_i$ is greater than $p_i$'s optimal match $\overline{\mathbf{u}}_i^m$, which is $\mu_{i,a_l^m}^m > \max\limits_{a_j^m \in \overline{\mathbf{u}}_i^m} \mu_{i,j}^m$.*

*Then the expected number of matches between $p_i$ and $a_l^m$ is upper bounded by*

$$\mathbb{E}[\Upsilon_{i,l}^m(T)] \le \min_{S^m \in \mathcal{C}(\mathcal{H}_{i,l}^m)} \sum_{(p_j, a_k^m, a_{k'}^m) \in S^m} \left( C_{i,j,k'}^m(T) + \frac{\log(T)}{d(\mu_{j,\overline{\mathbf{u}}_{i,\min}^m}, \mu_{j,k'})} \right),$$

*where $\overline{\mathbf{u}}_{i,\min}^m = \underset{a_k^m \in \overline{\mathbf{u}}_j^m}{\arg\min} \mu_{i,k}^m$, and $C_{i,j,k'}^m = \mathcal{O}((\log(T))^{-1/3})$.*

Then we provide the benefit (lower bound of the regret) of Machiavelli firm $p_i$ can gain by not following the MMTS from matching with $m$-type workers. Let's define the *super worker reward gap* as $\overline{\Delta}_{i,l}^m = \max\limits_{a_j^m \in \overline{\mathbf{u}}_i^m} \mu_{i,j}^m - \mu_{i,l}^m$, where $a_l^m \notin \overline{\mathbf{u}}_i^m$.

**Theorem G.1.** *Suppose all firms other than firm $p_i$ submit preferences according to the MMTS to the centralized platform. Then the following upper bound on firm $p_i$'s optimal regret for $m$-type workers*

*holds:*

$$R_i^m(T, \theta) \geq \sum_{l: \overline{\Delta}_{i,l}^m < 0} \overline{\Delta}_{i,l}^m \left[ \min_{S^m \in \mathcal{C}(\mathcal{H}_{i,l}^m)} \sum_{(p_j, a_k^m, a_{k'}^m) \in S^m} \left( C_{i,j,k'}^m + \frac{\log(T)}{d(\mu_{j, \overline{\mathbf{u}}_{i,\min}^m}, \mu_{j,k'})} \right) \right] \quad \text{(C.18)}$$

*where* $\overline{\mathbf{u}}_{i,\min}^m = \underset{a_k^m \in \overline{\mathbf{u}}_j^m}{\operatorname{argmin}} \mu_{i,k}^m$, *and* $C_{i,j,k'}^m = \mathcal{O}((\log(T))^{-1/3})$.

This result can be directly derived from Lemma G.1. Theorem G.1 demonstrates that there is no sequence of preferences that a firm can submit to the centralized platform that would result in negative optimal regret greater than $\mathcal{O}(\log T)$ in magnitude within type $m$. When considering multiple types together for firm $p_i$, this magnitude remains $\mathcal{O}(\log T)$ in total. Theorem G.1 confirms that, when there is a unique stable matching in type $m$, firms cannot gain significant advantage in terms of firm-optimal stable regret by submitting preferences other than those generated by the MMTS algorithm. An example is provided in Section 5.1 to illustrate this incentive compatibility property. Figure 2(b) illustrates the total regret, with solid lines representing the aggregate regret over all types for each firm, and dashed lines representing the regret for each type. It is observed that the type 1 regret of firm 1 is negative, owing to the inaccuracies in the rankings submitted by both firm 1 and firm 2. A detailed analysis of this negative regret pattern is given in Section I.2.

## G.1 PROOF OF INCENTIVE COMPATIBILITY

**Lemma G.2.** *Let* $\Upsilon_{i,l}^m(T)$ *be the number of times a firm* $p_i$ *matched with a* $m$-*type worker such that the mean reward of* $a_l^m$ *for firm* $p_i$ *is greater than* $p_i$*'s optimal match* $\overline{u}_i^m$, *which is* $\mu_{i,a_l^m}^m > \max_{a_j^m \in \overline{u}_i^m} \mu_{i,j}^m$.

*Then*

$$\mathbb{E}[\Upsilon_{i,l}^m(T)] \leq \min_{S^m \in \mathcal{C}(\mathcal{H}_{i,l}^m)} \sum_{(p_j, a_k^m, a_{k'}^m) \in S^m} \left( C_{i,j,k'}^m(T) + \frac{\log(T)}{d(\mu_{j, \overline{u}_{i,\min}^m}, \mu_{j,k'})} \right) \quad \text{(C.19)}$$

*where* $\overline{u}_{i,\min}^m = \underset{a_k^m \in \overline{u}_j^m}{\operatorname{argmin}} \mu_{i,k}^m$, $C_{i,j,k'}^m = \mathcal{O}((\log(T))^{-1/3})$.

*Proof.* We claim that if firm $p_i$ is matched with a *super optimal* $m-$ type worker $a_l^m$ in any round, the matching $u^m$ must be unstable according to true preferences from both sides. We then state that there must exist a $m$-type blocking triplet $(p_j, a_k^m, a_{k'}^m)$ where $p_j \neq p_i$.

We prove it by contradiction. Suppose all blocking triplets in matching $u$ *only* involve firm $p_i$ within $m-$ type worker. By Theorem 4.2 in (Abeledo and Rothblum, 1995), we can start from any matching $u$ to a stable matching by iteratively satisfying blocking pairs in a *gender consistent* order, which means that we can provide a well-defined order to determine which blocking triplet should be satisfied (matched) first within preferences from firm $p_i$[6]. Doing so, firm $p_i$ can never get a worse match than $a_l^m$ since a blocking pair will let firm $p_i$ match with a better $m-$ type worker than $a_l^m$, or become unmatched as the algorithm proceeds, so the matching will remain unstable. The matching will continue, which is a contradiction.

Hence there must exist a firm $p_j \neq p_i$ such that $p_j$ is part of a blocking triplet in $u$ when firm $p_i$ is matched with $m-$ type worker $a_l^m$ under the matching $u$. In particular, based on the Theorem 9 (Dubins-Freedman Theorem), firm $p_j$ must submit its TS preference.

Let $L_{j,k,k'}^m(T)$ be the number of times firm $p_j$ matched with $m-$ type worker $a_{k'}^m$ when the triplet $(p_j, a_k^m, a_{k'}^m)$ is blocking the matching provided by the centralized platform. Then by the definition

$$\sum_{u^m \in B_{j,k,k'}^m} \Upsilon_{u^m}(T) = L_{j,k,k'}^m(T) \quad \text{(C.20)}$$

By the definition of a blocking triplet, we know that if $p_j$ is matched with $m-$ type worker $a_{k'}^m$ when the blocking triplet $(p_j, a_k^m, a_{k'}^m)$ is blocking, the TS sample must have a higher mean reward for $a_{k'}^m$

---

[6]This gender consistent requirement is to satisfy a blocking pair $(p_j, a_k^m)$ and those blocking pairs can be ordered before we break their current matches if any, and then match $p_j$ and $a_k^m$ to get a new matching.

than $a_k^m$. In other words, we need to bound the expected number of times that the TS mean reward for $m-$ type worker $a_{k'}^m$ is greater than $a_k^m$. From (Komiyama et al., 2015), we know that the number of times that $(p_j, a_k^m, a_{k'}^m)$ forms a blocking pair in Thompson sampling, is upper bounded by

$$\mathbb{E} L_{j,k,k'}^m \leq C_{i,j,k'}^m(T) + \frac{\log(T)}{d(\mu_{j,\overline{u}_{i,\min}^m}, \mu_{j,k'})} \tag{C.21}$$

where $\overline{u}_{i,\min}^m = \underset{a_k^m \in \overline{u}_j^m}{\operatorname{argmin}} \mu_{i,k}^m$ and $C_{i,j,k'}^m = \mathcal{O}((\log(T))^{-1/3})$. The $d(x, y) = x \log(x/y) + (1 - x)\log((1-x)/(1-y))$ is the KL divergence between two Bernoulli distributions with expectation $x$ and $y$.

The expected number of times $\Upsilon_{i,l}^m(T)$ a firm $p_i$ matched with a $m-$ type worker such that the mean reward of $a_l^m$ for firm $p_i$ is greater than $p_i$'s optimal match $\overline{u}_i^m$, which is equivalent to the expected number of times viat the achievable sub-matching set $\Upsilon_{u^m}(T)$ where $u^m \in \mathcal{H}_{i,l}^m$. So the result then follows from the identity

$$\mathbb{E}[\Upsilon_{i,l}^m(T)] = \sum_{u^m \in \mathcal{H}_{i,l}^m} \mathbb{E}\Upsilon_{u^m}(T) \tag{C.22}$$

Given a set $\mathcal{H}_{i,l}^m$ of matchings, we say a set $S^m$ of triplets $(p_j, a_k^m, a_{k'}^m)$ is a *cover* of $\mathcal{H}_{i,l}^m$ if

$$\bigcup_{(p_j, a_k^m, a_{k'}^m) \in S^m} B_{j,k,k'}^m \supseteq H_{i,l}^m \tag{C.23}$$

Let $\mathcal{C}(H_{i,l}^m)$ denote the set of covers of $H_{i,l}^m$. Then

$$
\begin{aligned}
\mathbb{E}[\Upsilon_{i,l}^m(T)] &= \mathbb{E} \sum_{u^m \in \mathcal{H}_{i,l}^m} \Upsilon_{u^m}(T) \\
&\leq \mathbb{E} \min_{S^m \in \mathcal{C}(\mathcal{H}_{i,l}^m)} \sum_{(p_j, a_k^m, a_{k'}^m) \in S^m} \Upsilon_{u^m}(T) \\
&= \min_{S^m \in \mathcal{C}(\mathcal{H}_{i,l}^m)} \mathbb{E} \sum_{(p_j, a_k^m, a_{k'}^m) \in S^m} \Upsilon_{u^m}(T) \\
&= \min_{S^m \in \mathcal{C}(\mathcal{H}_{i,l}^m)} \sum_{(p_j, a_k^m, a_{k'}^m) \in S^m} \mathbb{E} L_{j,k,k'}^m(T) \\
&\leq \min_{S^m \in \mathcal{C}(\mathcal{H}_{i,l}^m)} \sum_{(p_j, a_k^m, a_{k'}^m) \in S^m} \left( C_{i,j,k'}^m(T) + \frac{\log(T)}{d(\mu_{j,k}, \mu_{j,k'})} \right) \\
&\leq \min_{S^m \in \mathcal{C}(\mathcal{H}_{i,l}^m)} \sum_{(p_j, a_k^m, a_{k'}^m) \in S^m} \left( C_{i,j,k'}^m(T) + \frac{\log(T)}{d(\mu_{j,\overline{u}_{i,\min}^m}, \mu_{j,k'})} \right)
\end{aligned}
\tag{C.24}
$$

where the first inequality is from the property of cover and we select the minimum cover $S^m$ from $\mathcal{C}(\mathcal{H}_{i,l}^m)$. And summation in the third line is equivalent to $\sum_{u^m \in B_{j,k,k'}^m}$. Based on Eq. (C.20), the third equality is obvious. From (Komiyama et al., 2015), we know the expected number of times of matching with the sub-optimal $m-$ type worker is upper bounded by Eq. (C.21). $\qquad\square$

## H  FIRM DA ALGORITHM WITH TYPE AND WITHOUT TYPE CONSIDERATION

In this section, we present the DA algorithm with type consideration and without type consideration.

---

**Algorithm 4:** Firm DA Algorithm with type.

| | |
|---|---|
| **Input** | : Type. firms set $\mathcal{N}$, workers set $\mathcal{K}_m, \forall m \in [M]$; firms to workers' preferences $\mathbf{r}_i^m, \forall i \in [N], \forall m \in [M]$, workers to firms' preferences $\boldsymbol{\pi}^m, \forall m \in [M]$; firms' type-specific quota $q_i^m, \forall i \in [N], \forall m \in [M]$, firms' total quota $Q_i, \forall i \in [N]$. |
| **Initialize** | : Empty set $\mathcal{S} = \{\}$, empty sets $S^m = \emptyset, \forall m \in [M]$. |

1 **for** $m = 1, ..., M$ **do**

2    **while** ∃ *A firm $p$ who is not fully filled with the quota $q^m$ and has not contacted every $m - type$ worker* **do**

3       Let $a$ be the highest-ranking worker in firm $p$'s preference, to whom firm $p$ has not yet contacted.

4       Now firm $p$ contacts the worker $a$.

5       **if** *Worker $a$ is free* **then**

6          $(p, a)$ become matched (add $(p, a)$ to $S^m$).

7       **else**

8          Worker $a$ is matched to firm $p'$ (add $(p', a)$ to $S^m$).

9          **if** *Worker $a$ prefers firm $p'$ to firm $p$* **then**

10             firm $p$ filled number minus 1 (remove $(p, a)$ from $S^m$).

11          **else**

12             Worker $a$ prefers firm $p$ to firm $p'$.

13             firm $p'$ filled number minus 1 (remove $(p', a)$ from $S^m$).

14             $(p, a)$ are paired (add $(p, a)$ to $S^m$).

15    Update: Add $S^m$ to $\mathcal{S}$.

| | |
|---|---|
| **Output** | : Matching result $\mathcal{S}$. |

---

**Algorithm 5:** Firm DA Algorithm without type (Gale and Shapley, 1962).

| | |
|---|---|
| **Input** | : Worker Types, firms set $\mathcal{N}$, workers set $\mathcal{K}_m, \forall m \in [M]$; firms to workers' preferences $\mathbf{r}_i^m, \forall i \in [N], \forall m \in [M]$, workers to firms' preferences $\boldsymbol{\pi}^m, \forall m \in [M]$; firms' type-specific quota $q_i^m, \forall i \in [N], \forall m \in [M]$, firms' total quota $Q_i, \forall i \in [N]$. |
| **Initialize** | : Empty set $S$. |

1 **while** ∃ *A firm $p$ who is not fully filled with the quota $\tilde{Q}$ and has not contacted every worker* **do**

2    Let $a$ be the highest-ranking worker in firm $p$'s preference over all types of workers, to whom firm $p$ has not yet contacted.

3    Now firm $p$ contacts the worker $a$.

4    **if** *Worker $a$ is free* **then**

5       $(p, a)$ become matched (add $(p, a)$ to $S$).

6    **else**

7       Worker $a$ is matched to firm $p'$ (add $(p', a)$ to $S$).

8       **if** *Worker $a$ prefers firm $p'$ to firm $p$* **then**

9          firm $p$ filled number minus 1 (remove $(p, a)$ from $S$).

10       **else**

11          Worker $a$ prefers firm $p$ to firm $p'$.

12          firm $p'$ filled number minus 1 (remove $(p', a)$ from $S$).

13          $(p, a)$ are paired (add $(p, a)$ to $S$).

| | |
|---|---|
| **Output** | : Matching result $S$. |

---

# I EXPERIMENTAL DETAILS

In this section, we provide more details about the analysis of the negative regret, parameters, and large market.

## I.1 NEGATIVE REGRET PHENOMENON

The occurrence of negative regret in multi-agent matching schemes presents an interesting phenomenon, contrasting the single-agent bandit problem wherein negative regret is non-existent.

Table 1: True matching rewards of two types of workers from two firms.

| Mean ID | Type | 1 | 2 | 3 | 4 | 5 |
|---|---|---|---|---|---|---|
| $\boldsymbol{\mu}_1$ | 1 | 0.406 | 0.956 | 0.738 | 0.970 | 0.695 |
| | 2 | 0.932 | 0.241 | 0.040 | 0.657 | 0.289 |
| $\boldsymbol{\mu}_2$ | 1 | 0.682 | 0.909 | 0.823 | 0.204 | 0.218 |
| | 2 | 0.303 | 0.849 | 0.131 | 0.886 | 0.428 |

Table 2: Estimated mean reward and variance of each type of worker in view of two firms. The bold font is to represent the firm's optimal stable matching. † represents the difference between the estimated mean and the true mean less than $1\%$. ‡ represents the difference is less than $1.5\%$.

| Mean & Var | Type | 1 | 2 | 3 | 4 | 5 |
|---|---|---|---|---|---|---|
| $\hat{\boldsymbol{\mu}}_1$ | 1 (DS) | $0.533_{0.015}$ | $\mathbf{0.943}^{\ddagger\dagger}_{0.000}$ | $0.917_{0.035}$ | $\mathbf{0.968}^{\dagger}_{0.000}$ | $0.682^{\ddagger}_{0.003}$ |
| | 2 (SDE) | $\mathbf{0.950}_{0.000}$ | $0.223_{0.000}$ | $0.041^{\dagger}_{0.000}$ | $0.500_{0.208}$ | $\mathbf{0.303}^{\ddagger}_{0.000}$ |
| $\hat{\boldsymbol{\mu}}_2$ | 1 (DS) | $\mathbf{0.683}^{\dagger}_{0.000}$ | $0.500_{0.035}$ | $\mathbf{0.823}^{\dagger}_{0.000}$ | $0.262_{0.037}$ | $\mathbf{0.210}^{\dagger}_{0.000}$ |
| | 2 (SDE) | $0.083_{0.035}$ | $\mathbf{0.851}^{\dagger}_{0.000}$ | $0.124^{\dagger}_{0.001}$ | $\mathbf{0.887}^{\dagger}_{0.000}$ | $0.415^{\ddagger}_{0.001}$ |

In the context of the single-agent bandit problem, it is known that the best arm can be pulled, resulting in instantaneous regret that can attain zero but not take negative values. Conversely, in the multi-agent competing bandit problem, the oracle firm-optimal arm is determined by the true expected reward/utility, assuming knowledge of the true parameter $\mu^*$. However, due to the imprecise estimation of rankings/parameters at each time step, an exact match with the oracle policy cannot be guaranteed. This discrepancy leads to varied outcomes for firms in terms of benefits (negative instantaneous regret) or losses (positive instantaneous regret) from the matching process. Instances arise where firms may strategically submit inaccurate rankings to exploit these matches, a phenomenon termed machiavelli/strategic behaviors. Nevertheless, over the long term, such strategic actions do not yield utility gains in accordance with our policy.

Furthermore, it is crucial to note that our matching solution remains a stable matching at each time step. This means that the stable matching remains independent of the negative regret generated by our policy, as stable matching is a short-term discrete metric, while regret serves as a long-term evaluation continuous metric.

## I.2 LEARNING

In this section, we present the learning parameters of $(\boldsymbol{\alpha}, \boldsymbol{\beta})$ of Example 1. Besides, we analyze which kind of pattern causes the non-optimal stable matching of Examples 1 and 2.

**Findings from Example 1.**

We show the posterior distribution of $(\boldsymbol{\alpha}, \boldsymbol{\beta})$ in Figure 3. The first and second row represents the posterior distributions of firm 1 and firm 2 over two types of workers after $T$ rounds interaction. The first and second columns in Figure 3 represent two firms' posterior distributions over type I and type II workers.

We find that the posterior distributions of the workers that firms most frequently match with exhibit a relatively sharp shape, indicating that firms can easily construct uncertainty sets over these workers. However, in some instances, the distributions are relatively flat, indicating a lack of exploration. This can be attributed to two possible reasons: (1) the workers in question are not optimal stable matches for the firms, and are thus abandoned early on in the matching process, such as firm 1's DS 1 and DS 5, or (2) the workers are optimal, but are erroneously ranked by the firms and subsequently blocked, such as firm 2's SDE 3. To further illustrate this, we present the posterior mean and variance in Table 2. The optimal stable matches for each firm are represented in bold, and the variance of the distributions is denoted by small font. Additionally, we use the dagger symbol to indicate when the difference between the posterior mean reward and true matching reward is less than $1\%$ and $1.5\%$.

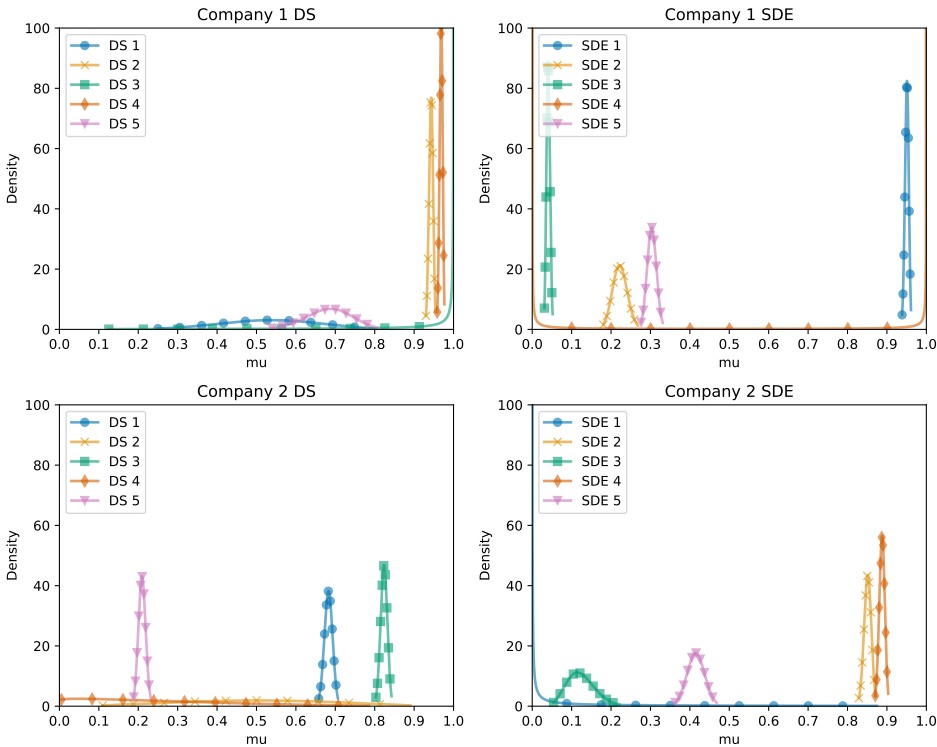

Figure 3: Posterior distribution of learning parameters for two firms in Example 1.

**Pattern Analysis.** We find that firm 1's type I matching in Figure 2(b), achieves a negative regret due to the high-frequency matching pattern of $\mathbf{u}_1 = \{[D_4, D_2, D_5], [S_1, S_5]\}$, and $\mathbf{u}_2 = \{[D_3, D_1], [S_4, S_2, S_3]\}$. That means firm 1 and firm 2 have a correct (stable) matching in the first match $\tilde{\mathbf{u}}_1 = \{[D_4, D_2], [S_1, S_5]\}, \tilde{\mathbf{u}}_2 = \{[D_3, D_1], [S_4, S_2]\}$. In the second match, they both need to compare worker $D_5$ and worker $S_3$, because all other workers are matched with firms or have been proposed in the first match. In Table 1, we find that two workers' true mean rewards for firm 1 are $\mu_{1,5}^1 = 0.695, \mu_{1,3}^2 = 0.040$ and two workers' estimated rewards for firm 1 are $\hat{\mu}_{1,5}^1 = 0.682, \hat{\mu}_{1,3}^2 = 0.041$. These two workers are pretty different and can be easily detected. So firm 1 has a high chance of ranking them correctly. However, two workers' true rewards for firm 2 are $\mu_{2,5}^1 = 0.218, \mu_{2,3}^2 = 0.131$, and two workers' estimated rewards for firm 2 are $\hat{\mu}_{1,5}^1 = 0.210, \hat{\mu}_{1,3}^2 = 0.124$. These workers are close to each other, where these two posteriors' distributions overlap a lot and can be checked in Figure 3. So firm 2 has a non-negligible probability to incorrectly rank $S_3$ ahead of $D_5$. Therefore, based on the true preference, firm 2 could match with $S_3$ and firm 1 matches with $D_5$ with a non-negligible probability rather than the optimal stable matching $(p_1, S_3)$ and $(p_2, D_5)$ by $D_5$ preferring firm 2.

The above pattern links to Section 3.2, incapable exploration, and Section G, incentive compatibility. Due to the insufficient exploration of $S_3$ and $D_5$, firm 2 may rank them incorrectly to get a match with $S_3$ rather than optimal $D_3$ and the regret gap is $\mu_{2,3}^1 - \mu_{2,3}^2 = 0.823 - 0.131 = 0.692$, which is a positive instantaneous regret. Due to the incorrect ranking from firm 2, firm 1 gets a final match with $D_5$ rather than optimal $S_3$, and suffers a regret gap $\mu_{1,3}^2 - \mu_{1,5}^1 = 0.040 - 0.695 = -0.655$, which is a negative instantaneous regret. Thus firm 1 benefits from firm 2's incorrect ranking and can achieve a total negative regret, as shown in Figure 2(b).

**Findings from Example 2.** In our analysis of the non-optimal stable matching in Example 2, we observed that both firms incurred positive total regret, shown in Figure 2(c). We find that the quota setting resulted in all workers of type II being assigned to firms in the first match. As a result, in the second match, the ranking submitted by firm 1 to the centralized platform did not affect firm 2's matching result for type II workers. This can be thought of as an analogy where firms are schools and workers are students. In the second stage of the admission process, school 2 would not participate in

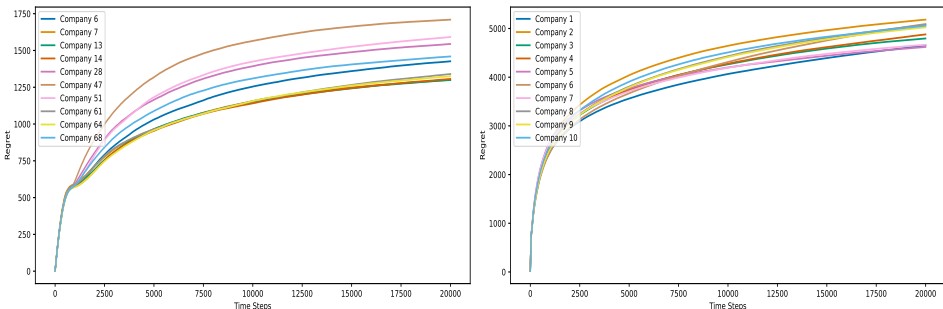

Figure 4: Left: 10 out of 100 randomly selected firms' total regret in Examples 3. Right: all firms' total regret in Example 4.

the competition for type II students, and its matching outcome would not be affected by the strategic behavior of other schools in the second stage, but rather by the strategic behavior of other schools in the first stage.

### I.3 LARGE MARKETS

In this part, we provide two large market examples to demonstrate the robustness of our algorithm. All preferences are randomly generated and all results are over 50 trials to take the average.

**Example 3.** We consider a large market composed of many firms ($N = 100$) and many workers ($K_1 = K_2 = 300$). Besides, we have $Q_1 = Q_2 = 3, q_1^1 = q_2^1 = q_2^1 = q_2^2 = 1$.

**Example 4.** We also consider a large market consisting of many workers, and each firm has a large, specified quota and an unspecified type quota. In this setting, $N = 10, M = 2, K_1 = K_2 = 500, Q_1 = Q_2 = 30, q_1^1 = q_2^1 = q_2^1 = q_2^2 = 10$.

**Results.** In Figure 4(a), we randomly select 10 out of 100 to present firms' total regret, and all those firms suffer sublinear regret. In Figure 4(b), we also show all 10 firms' total regret. Comparing Examples 3 and 4, we find that firms' regret in Example 3 is less than firms' regret from Example 4 because in Example 4, each firm has more quotas (30 versus 3), which demonstrates our findings from Theorem 4.2. In addition, we find there is a sudden exchange in Figure 4(a) nearby time $t = 1500$. We speculate this phenomenon is due to the small gap between different workers and the shifting of the explored workers.

## J ADDITIONAL RELATED WORKS

**Multi-Agent Systems and Game theory.** There are some papers considering the multi-agent in the sequential decision-making systems including the cooperative setting (Littman, 2001; González-Sánchez and Hernández-Lerma, 2013; Zhang et al., 2018; Perolat et al., 2018; Shi et al., 2022) and competing setting (Littman, 1994; Auer and Ortner, 2006; Zinkevich et al., 2007; Wei et al., 2017; Fiez et al., 2019; Jin et al., 2020). Zhong et al. (2021) study the multi-player general-sum Markov games with one of the players designated as the leader and the other players regarded as followers and establish the efficient RL algorithms to achieve the Stackelberg-Nash equilibrium.

**Assortment Optimization.** To maximize the number of matches between the two sides (customers and suppliers), the platform must balance the inherent tension between recommending customers more potential suppliers to match with and avoiding potential collisions. Ashlagi et al. (2022) introduce a stylized model to study the above trade-off. Motivated by online labor markets (Aouad and Saban, 2022) consider the online assortment optimization problem faced by a two-sided matching platform that hosts a set of suppliers waiting to match with a customer. Immorlica et al. (2021) consider a two-sided matching assortment optimization under the continuum model and achieve the optimized meeting rates and maximize the equilibrium social welfare. Rios et al. (2022) discuss

the application of assortment optimization in dating markets. Shi (2022) studies the minimum communication needed for a two-sided marketplace to reach an approximately stable outcome with the transaction price.

**Matching Markets.** One strand of related literature is two-sided matching, which is a stream of papers that started in (Gale and Shapley, 1962). They propose the deferred acceptance (DA) algorithm (also known as the GS algorithm) with its application in the marriage problem and college admission problem. A series work (Knuth, 1976; Roth, 1982; Roth and Sotomayor, 1992; Roth, 2008) discuss the history of the DA algorithm and summarize theories about stability, optimality, and incentive compatibility, and finally provide its practical use and further open questions. In particular, Roth (1985); Sönmez (1997) propose that the college admissions problem is not equivalent to the marriage problem, especially when a college can manipulate its capacity and preference. Notably, in the hospital doctor matching example, since hospitals want diversity of specializations, or demographic diversity, or whatever, they care about the combination (group of doctors) they get. Roth (1986) state that when all preferences are strict, and hospitals (firms) have responsive preferences, the set of doctors (workers) employed and positions filled is the same at every stable match. However, when there exist *couples* in the preference list (not *responsive preference* (Klaus and Klijn, 2005)), which might make the set of stable matchings empty. Even when stable matchings exist, there need not be an optimal stable matching for either side. Later, Ashlagi et al. (2011) revisit this couple matching problem and provide the *sorted deferred acceptance algorithm* that can find a stable matching with high probability in large random markets. Biró et al. (2014) provide an integer programming model for hospital/resident problems with couples (HRC) and ties (HRCT). Manlove et al. (2017) release the HRC with minimal blocking pairs and show that if the preference list of every single resident and hospital is of length at most 2, their method can find a polynomial-time algorithm. Nguyen and Vohra (2018; 2022) find the stable matching in the nearby NRC problem, which is that the quota constraints are soft. Azevedo and Hatfield (2018); Che et al. (2019); Greinecker and Kah (2021) discuss the existence and uniqueness of stable matching with complementaries and its relationship with substitutable preferences in large economies. Besides, there are also papers considering stability and optimality of the refugee allocation matching (Aziz et al., 2018; Hadad and Teytelboym, 2022). Tomoeda (2018); Boehmer and Heeger (2022) consider a case that firms have hard constraints both on the minimum and maximum type-specific quotas and other type-specific quota consideration works.