# OpenReview forum: "Two-sided Competing Matching Markets With Complementary Preferences"
_ICLR.cc/2024/Conference — ICLR 2024 Conference Withdrawn Submission_

### Official Review · Reviewer_ULou · 2023-10-14

**Soundness:** 2 fair
**Presentation:** 2 fair
**Contribution:** 2 fair
**Rating:** 3
**Confidence:** 3

**Summary:**

The paper studies matching markets with complementaries in an online learning framework. The model is as follows: a set of firms want to hire a set of workers; workers and firms have mutual preferences, and the workers have types. Each firm has a quota of each worker’s type to fill and can be allocated at most a certain amount of workers, so there are complementarities between the workers the firm hires.
While the ranking of firms’ preferences is publicly known, the corresponding valuations and the workers’ preferences are not. A centralized platform (the learner) repeatedly allocates workers to firms to maximize the value of the allocation for the firms while enforcing stability; at the end of each round, the centralized platform sees the realized value for the firms (bandit feedback).

The main contribution of the paper is a learning algorithm that achieves sublinear regret with respect to the best-fixed allocation in hindsight while enforcing stability and some notion of incentive compatibility.

**Strengths:**

Online learning has been applied with vast success to economic models, and, besides the theoretical relevance, it is well-motivated in practice.

**Weaknesses:**

My main concern, which motivates my low score, concerns the model. Furthermore, I find the discussion on incentive compatibility (postponed to the appendix) not satisfactory.

(a) it is unclear to me why we need learning at all: the vanilla deferred acceptance mechanism is truthful on the workers’ side, meaning that the workers have no incentive to lie. It is not clear to me whether this is the case for the matching algorithm (Algorithm 3) used by the authors. There are two options: either this new mechanism is truthful for the workers, in which case the platform should ask the workers about their preferences, with no learning involved, or it is not, in which case the workers may not respond truthfully to the queries of the learning algorithm. In both cases, there is an issue with the model.

(b) Concerning firms’ valuations, where are they used? Algorithms 4 and 5 are based on the rankings and make no use of the valuations, so what is the point of learning valuations? Also, it is suspicious that an algorithm agnostic to the valuations manages to perform, on average, close to the optimal stable matching, given that many stable matching may exist.

(c) Concerning truthfulness, what the authors prove is not an incentive compatibility result, as incentive compatibility means that agents have no incentive to deviate from truth-telling, regardless of what other agents do. The authors prove some kind of equilibrium result that says that truthtelling is a Nash equilibrium up to a lower-order term. This latter result is weaker and does not guarantee truthfulness.

**Questions:**

See points (a) to (c) above.

---

### Official Review · Reviewer_YP85 · 2023-10-31

**Soundness:** 3 good
**Presentation:** 3 good
**Contribution:** 3 good
**Rating:** 5
**Confidence:** 3

**Summary:**

This paper introduces the problem formulation of Competing Matching under Complementary Preferences problem (CMCP), which matches agents with arms. The considered setting is a many-to-one scenario where one agent can be matched with multiple arms. The authors proposes a centralized algorithm: MMTS, and provide regret analysis of the algorithm.

**Strengths:**

Strength:
1. The problem formulation of the Competing Matching under Complementary Preferences problem (CMCP) is interesting. The new algorithm MMTS using double matching is the first algorithm to achieve stability and IC in the CMCP.
2. The presentation is good, and the paper is easy to follow.

**Weaknesses:**

Weakness:
1. It seems the major contributes is on the formulation of CMCP into a bandit learning framework, with relatively standard analysis technique.
2. Assumptions: The authors consider the marginal preferences (MP) rather than joint (couple) preferences (JP), more justifications could be helpful.
3. The paper discusses the centralized setting, which may not be practical, as in the motivating examples provided by the authors. With these strong assumptions, the technical novelty is limited.

**Questions:**

See weakness.

---

### Official Review · Reviewer_vqtQ · 2023-11-05

**Soundness:** 3 good
**Presentation:** 3 good
**Contribution:** 3 good
**Rating:** 6
**Confidence:** 2

**Summary:**

This paper proposes a new algorithm called Multi-agent Multi-type Thompson Sampling (MMTS) for addressing the challenge of matching markets with complementary preferences. Complementary preferences occur when agents on one side of the market prefer to be matched with agents on the other side who have certain characteristics or attributes. For example, in a job market, employers may prefer to hire employees with certain skills or qualifications, while employees may prefer to work for employers who offer certain benefits or work culture.

The MMTS algorithm combines the strengths of Thompson Sampling for exploration with a double matching technique to achieve a stable matching outcome. The algorithm works by modeling the preferences of agents on both sides of the market as probability distributions, and then using Thompson Sampling to sample from these distributions and explore the space of possible matches. The double matching technique ensures that the final matching outcome is stable, meaning that there are no two agents who would prefer to be matched with each other instead of their current matches.

The paper also presents simulation results to demonstrate the effectiveness of MMTS in learning the unknown preferences of firms. The results show that MMTS is able to achieve a low regret, meaning that it makes few mistakes in matching agents with their preferred partners. The paper also discusses the computational complexity of MMTS, as well as its stability properties and regret upper bound.

**Strengths:**

The authors proposed an interesting variant of two-sided matching markets when the participants have complementary preferences.

The authors designed the Multi-agent Multi-type Thompson Sampling (MMTS) algorithm which builds on the strengths of Thompson Sampling in terms of exploration and a double matching technique to find a stable solution.

The writing is relatively acceptable.

**Weaknesses:**

I am not an expert in this area and provide the following suggestions, mostly on writings, for the authors’ consideration. These are not necessarily weaknesses of the paper.

1.	I am not sure if the title (complementary preferences) is proper as the authors actually study the matching with upper and lower quotas.

2.	The introduction section can be expanded and better motivated.

The model with complementary preferences is only motivated in the first paragraph of the introduction with one sentence of examples. I would recommend the authors expand this paper, possibly starting with an intuitive and informal definition of complementary preferences, followed by these real-world examples with detailed explanations.

I think the authors also need to explain why these real-world problems can be modeled as a bandit learning framework.

The authors claim that “Matching markets often consist of participants with complementary preferences that can lead to instability (Che et al., 2019).” I do not understand this sentence, as I think the “instability” is caused by the algorithm instead of the model.

3.	I have some problems with the problem section.

The authors claim that the time horizon T can be assumed to be known without loss of generality using the doubling trick (Auer et al., 1995). I am not familiar with this trick and a short explanation may be helpful, but the authors can ignore this comment if this assumption is common and the technique is well-known.

The assumption of strict preferences can be explained in detail. The authors explained that strict preference is not necessary for stable matching and regret metric, and for simplicity, the assumption is made. In the conclusion section, the scenarios where agents have indifferent preferences are left for future work. I think the open problems can be made more specific, and the effect of indifferent preferences can be better highlighted.

I think it looks neat if “$m - $ type” is written as “$m$-type”.

4.	The sections of algorithms and properties:

It is good that the authors provide explanations of the algorithms which are helpful for me to understand the design. However, due to space limit, Algo 4 and 5 can only be put in the appendix. To understand the main algorithms, since the authors actually need to know the properties of these algorithms (e.g., each firm will not propose to the previous workers who rejected him/her in Algo 5), I am wondering if it is possible to include a short description of how these algorithms run.

For me, I think it is a quite good result that the double matching technique can provide stable matching solutions in the complete information setting, although the algorithm does not provide much technical contribution. I am also surprised this is not known in the literature.

I did not check the proof of Theorem 4.1.

The writing can be consistent. For example, Page 5, “Step 2: Double Matching Stage” and Page 3, “Definition 1 (Blocking Pair)”.

5.	Related work. I personally think this section can be expanded (some materials in the appendix can be mentioned in the main body). I guess the audience may want to know for what variants, the existence and computational complexity of stable matchings are known.

**Questions:**

I do not have specific questions. Perhaps the authors can explain the effect of indifferent preferences to their model.

---

### Official Review · Reviewer_cZ2J · 2023-11-05

**Soundness:** 2 fair
**Presentation:** 1 poor
**Contribution:** 2 fair
**Rating:** 3
**Confidence:** 4

**Summary:**

This paper deals with a variant of the two sided many-to-one multi-type matching problem with complementary preferences. In their setting there are two sides: firms and workers. There are different types of workers. Each worker has publicly known preferences over the firms. Each firm has a minimum quota (demand) for each type of work which must be fulfilled and a maximum quota of total number of workers matched to it. Each firm has complementary preferences over the workers. The main challenge is that the firms' preferences are unknown. The goal is to learn firms' preferences over a publicly known time horizon T and compute firm optimal (regret minimizing) stable matchings. In each round t, each firm obtains a noisy sample of each agent's utility to the firm from a Beta distribution using learned parameters. The firm submits the rankings over workers of each type that these induce to a mechanism (a variant of the deferred acceptance algorithm) which computes a matching from which each firm derives a reward, and updates its belief about the parameters of the distribution from which the utilities of the agents to the firm are drawn. Here, the reward experienced by the firm from being matched to a worker is the true utility the worker would generate with some randomly generated noise added to it.

**Strengths:**

- One of the main conceptual contributions of this paper is the problem setting of two sided multi-type matching with complementary preferences.
- The paper makes some interesting modeling choices such as having minimum per-type quotas and maximum total quotas for firms and complimentary preferences.
- The results on the stability and regret bounds on the proposed multi-type multi-agent Thompson sampling (MMTS) algorithm appears to be sound, albeit under what seems like significant restrictions or assumptions.
- The topic of the paper is clearly relevant to ICLR and will likely be of interest to the community of computational social choice and matching researchers.

**Weaknesses:**

- I'm not sure about the motivation for some aspects of the problem setting and how the MMTS algorithm is defined to address it. E.g. in each round of MMTS, in Step 1, it seems like every firm samples the utility that every worker generates to the firm. Is this correct? Further, agents seem to report these utilities and induced rankings to the double matching stage (Step 2). Is this correct? Are the reports made in a way to strategically maximize the firm's reward? I'm unsure of what the role of Thompson sampling is in this context.
- There are several missing details and inconsistencies in the notation due to which I am unsure about the significance and soundness of some of the claims in the paper. I will provide some examples below, although they are not exhaustive.
- It is not clear to me how preferences are modeled exactly.
  - First, there seem to be some typos in the notation. E.g. from a strict reading of the notation in Section 2.1 parts II a and II b, a worker's preferences is given by a function that maps an agent to the set of firms. I believe the authors mean that each worker is mapped to a permutation over the firms, but the notion of permutations over firms are not formally introduced. Also, I am not familiar with the phrase "subset of the permutation", although I believe I understand what the authors mean. I hope the authors will consider a thorough rewriting of Section 2.1.
  - Please consider defining \mu_{i,j}^{m} in the first paragraph of Section 2.1 part II b.
  - I do not understand Figure 1. First, the notation used is not consistent with Section 2.1 part I. Next, it appears from the figure that the item b_1 of type 2 has a positive value for both \mu_i^1 and \mu_i^2. Does this mean b_1 provides agent i utilities for both types? In Fig 1, top row, left figure, what is the utility agent i gets from the matching {a_1, b_2}? What does the bidirectional edge between a_1 and a_2 represent? Aren't preferences strict as discussed earlier in the paper? Please also consider increasing the font size.
  - Section 2.1 part III. u_t^m(p_i) \in \K_m \cup \emptyset. Do you mean u_t^m(p_i) \subseteq \K_m \cup \emptyset since each firm can be matched to multiple workers of type m?
- Importantly it appears that the MMTS algorithm, and in particular, the output of the dual matching algorithm is only stable under the assumption that the firms' preferences can be marginalized, i.e. the preferences over workers of one type do not depend on the preferences over workers of other types that are matched to the firm. Am I missing something?
- Section 2.1 part V: Could you please define the optimal stable matching oracle policy \bar{u}_i^m?

**Questions:**

Please see the questions in the detailed comments above.